# Breast and pancreatic cancer interrupt IRF8-dependent dendritic cell development to overcome immune surveillance

Melissa A. Meyer[1], John M. Baer[1], Brett L. Knolhoff[1], Timothy M. Nywening[2], Roheena Z. Panni[1,2], Xinming Su[1], Katherine N. Weilbaecher[1,3], William G. Hawkins[2,3], Cynthia Ma[1,3], Ryan C. Fields[2,3], David C. Linehan[4], Grant A. Challen[1,3,5], Roberta Faccio[6], Rebecca L. Aft[2,3,7] & David G. DeNardo[1,3,8]

Tumors employ multiple mechanisms to evade immune surveillance. One mechanism is tumor-induced myelopoiesis, whereby the expansion of immunosuppressive myeloid cells can impair tumor immunity. As myeloid cells and conventional dendritic cells (cDCs) are derived from the same progenitors, we postulated that myelopoiesis might impact cDC development. The cDC subset, cDC1, which includes human CD141$^+$ DCs and mouse CD103$^+$ DCs, supports anti-tumor immunity by stimulating CD8$^+$ T-cell responses. Here, to understand how cDC1 development changes during tumor progression, we investigated cDC bone marrow progenitors. We found localized breast and pancreatic cancers induce systemic decreases in cDC1s and their progenitors. Mechanistically, tumor-produced granulocyte-stimulating factor downregulates interferon regulatory factor-8 in cDC progenitors, and thus results in reduced cDC1 development. Tumor-induced reductions in cDC1 development impair anti-tumor CD8$^+$ T-cell responses and correlate with poor patient outcomes. These data suggest immune surveillance can be impaired by tumor-induced alterations in cDC development.

---

[1] Department of Medicine, Washington University School of Medicine, St. Louis, MO 63110, USA. [2] Department of Surgery, Washington University School of Medicine, St. Louis, MO 63110, USA. [3] Siteman Cancer Center, Washington University School of Medicine, St. Louis, MO 63110, USA. [4] Department of Surgery, University of Rochester Medical Center, Rochester, NY 14642, USA. [5] Section of Stem Cell Biology, Division of Oncology, Washington University School of Medicine, St. Louis, MO 63110, USA. [6] Department of Orthopedic Surgery, Washington University School of Medicine, St. Louis, MO 63110, USA. [7] John Cochran St. Louis Veterans Administration Hospital, St. Louis, MO 63106, USA. [8] Department of Pathology and Immunology, Washington University School of Medicine, St. Louis, MO 63110, USA. Correspondence and requests for materials should be addressed to D.G.D. (email: ddenardo@wustl.edu)

To subvert immune surveillance, solid tumors disrupt tumor-targeted immune responses. Conventional dendritic cells (cDCs) support anti-tumor adaptive immunity by stimulating T cells, but cDCs often fail to accumulate in the tumor microenvironment[1,2]. Furthermore, those cDCs found in the tumor can be immature and are therefore less effective in antigen presentation and T-cell stimulation[3–6]. Solid tumors also interfere with anti-tumor immune responses by stimulating immature granulocyte and monocyte production from bone marrow (BM) progenitors. Expanded myeloid cells are recruited to tumors where they can maintain an immature phenotype or differentiate into tumor-associated macrophages. All of these populations can suppress anti-tumor CD8$^+$ T cells as well as promote tumor progression through support of angiogenesis and metastasis[7–10]. Interestingly, cDCs are produced from the same BM progenitors as the expanding populations of granulocytes and monocytes[11,12]. Although granulocyte and monocyte differentiation is known to be dysregulated in cancer[7–10], we do not fully understand how tumors affect cDC differentiation. Because of their common origin, we hypothesized that tumor-induced expansion of immature granulocytes and monocytes occur at the expense of cDC differentiation.

cDCs play an important role in initiating and maintaining adaptive immune responses. cDCs are split into two subsets: cDC1s, which specialize in CD8$^+$ T-cell activation, and cDC2s, which specialize in CD4$^+$ T-cell activation. The cDC1s are marked by CD141 in humans and encompass both migratory CD103$^+$ cDC1s and lymphoid-resident CD8α$^+$ cDC1s in mice. cDC2s are also found in both lymphoid and peripheral tissues and are marked by CD1c in humans and CD11b and Sirpα in mice[13–16]. The development of cDC1s is driven by the transcription factors interferon regulatory factor-8 (IRF8), basic leucine zipper transcription factor ATF-like 3 (Batf3), and inhibitor of DNA binding 2 (Id2)[13,17–19]. The development of cDC2s is driven by a different set of transcription factors including interferon regulatory factor 4 (IRF4).

Because of their role in activating CD8$^+$ T cells, cDC1s have been implicated in supporting the T-cell response against solid tumors. CD103$^+$ cDC1s are known to cross-present antigen to activate CD8$^+$ T cells and secrete factors that attract T cells into the tumor[18,20,21]. Furthermore, CD103$^+$ cDC1s are important for transporting antigen into the draining lymph nodes (LNs), supporting T-cell activation and expansion[2,22,23]. Given these functions, it is understandable that CD103$^+$ cDC1s have been implicated in initiation and maintenance of CD8$^+$ T-cell responses against tumors. CD103$^+$ cDC1s are required to restrain tumor growth and support response to CD8$^+$ T cell-mediated chemo- and immune-therapies in multiple mouse models of solid tumors. In patients, intratumoral CD141$^+$ cDC1 numbers correlate with better outcomes in many types of solid tumors, including breast cancer (BC)[1–3,18,24–26]. Thus, cDC1s are important mediators of the anti-tumor CD8$^+$ T-cell response and can function to control tumor progression in mice and humans.

Given their important role in supporting anti-tumor immunity, we inquired whether tumor progression affects the generation of cDCs. Recent work has shown that after committing to the granulocyte, monocyte, or cDC lineage, cDC precursors can commit to the cDC1 subset during differentiation before leaving the BM[27–31]. Given that differentiation choice can be made outside the tumor microenvironment, we hypothesized that systemic changes induced by tumors might impair cDC, and further cDC1, commitment in the BM, and subsequently influence cDC1s in the periphery and anti-tumor immunity. In this study, we show that tumors interrupt cDC, and specifically cDC1, differentiation in BC and pancreatic ductal adenocarcinoma (PDAC) mouse models and patients. This interruption reduces

the systemic cDC1 pool, negatively impacting CD8$^+$ T-cell immunity and correlating with poor patient outcome. Our data illustrate a new mechanism by which tumors subvert anti-tumor immunity via dysregulation of cDC1 differentiation.

## Results

**BC and PDAC patients have reduced BM cDC differentiation.** cDCs are important for initiating and sustaining anti-tumor T-cell responses[1,2,22]. To understand the impact of tumors on development of cDCs, we profiled BM samples from human BC and PDAC patients with localized disease and no prior therapy. We analyzed the following cDC progenitors: macrophage-dendritic cell progenitors (MDPs), which retain monocyte/macrophage potential; common dendritic cell progenitors (CDPs), which retain plasmacytoid DC (pDC) potential; and pre-DCs, which are committed to the cDC lineage. Relative to BM from healthy controls, BM from both BC and PDAC patients had decreased numbers of CDPs and pre-DCs. MDPs were reduced in BC patients but not PDAC patients (Fig. 1a, b, Supplementary Fig. 1a, Supplementary Table 1). These data suggest the defect in BM progenitors is strongest and most consistent after progenitors start to commit to the cDC lineage. We also analyzed cells of the cDC subsets, cDC1s and cDC2s, found in the BM. Patient populations showed a decrease in CD141$^+$ cDC1s and CD1c$^+$ cDC2s (Fig. 1a, b). As illustrated in Fig. 1a, CD141$^+$ cDC1s are rare in the BM but are dramatically reduced in BC and PDAC patients. We validated these findings using a second cohort of BC patients (Supplementary Fig. 1b). In contrast to the cDC lineage, an expansion of BM immature granulocytes (CD11b$^+$CD33$^{Hi}$CD14$^-$CD15$^+$), potentially a subset of myeloid-derived suppressor cells, was observed in both cancer types (Fig. 1b). This result is in agreement with published observations in several other solid tumor types[8,32–35]. To determine if the reduction in cDCs extended into the periphery, where cDCs are thought to function[13,18], we analyzed BC and PDAC patient blood samples and found reduced pre-DC numbers and increased immature granulocyte numbers relative to healthy controls (Fig. 1c, Supplementary Fig. 1c). We validated these findings in a second cohort of PDAC patients (Supplementary Fig. 1d). Together, these data suggest that cancer results in decreased cDC lineage cells in the BM and decreases in the systemic pool of pre-DCs.

Previous studies have shown that CD141$^+$ cDC1 numbers and functions in the tumor are predictive of patient outcome[1,3,36], prompting us to assess whether changes in CD141$^+$ cDC1s in the BM prior to treatment or resection were also predictive of response to therapy. We observed higher numbers of CD141$^+$ cDC1s and lower numbers of granulocytes (CD45$^+$CD11b$^+$CSF1R$^-$CD15$^+$CD14$^-$) in the BM of BC patients who achieved a pathological complete response (pCR) to neoadjuvant chemotherapy (Fig. 1d, Supplementary Table 1). Notably, CD1c$^+$ cDC2 numbers were not predictive of pCR, suggesting the predictive nature is specific to the CD141$^+$ cDC1 subset (Supplementary Fig. 1e). These findings demonstrate that tumor-induced decreases in CD141$^+$ cDC1 development in the BM could be an important indicator of patient immune competency and response to therapy.

**Local breast and pancreatic tumors systemically reduce cDC1s.** To more carefully study changes in cDC development during cancer progression, we evaluated three distinct genetic mouse models and four syngeneic orthotopic mouse models of BC and PDAC. Similar to the human cancer patients, we found six of these seven models had reductions in absolute numbers of BM CD24$^+$ cDC1s and most had reductions in the absolute number of BM pre-DCs (Fig. 2a–c, Supplementary Fig. 3a–d). To further

investigate the cDC lineage in tumor-bearing mice, we employed PyMT-B6, a cell line derived from the genetic MMTV-Polyoma Middle T (PyMT) mammary tumor mouse model on the C57BL/6 background. Consistent with the patient data, mice bearing end-stage orthotopic PyMT-B6 mammary tumors showed no change in MDPs but a modest reduction in CDPs (Fig. 2b, Supplementary Fig. 2a). This finding suggests that tumors impact

components of the cDC1 lineage. In contrast to cDC1s, Ly6G$^+$ Ly6C$^+$CSF1R$^+$ immature granulocytes and Ly6G$^+$CSF1R$^-$ mature granulocytes expand in the BM (Fig. 2b, Supplementary Fig. 2a). In addition, the changes in cDC progenitors and granulocytic populations were consistent in the genetic models of BC and PDAC: MMTV-PyMT FvB/N and KPC (p48-Cre;LSL-Kras$^{G12D}$;Trp53$^{flox/+}$ C57BL/6), respectively (Supplementary

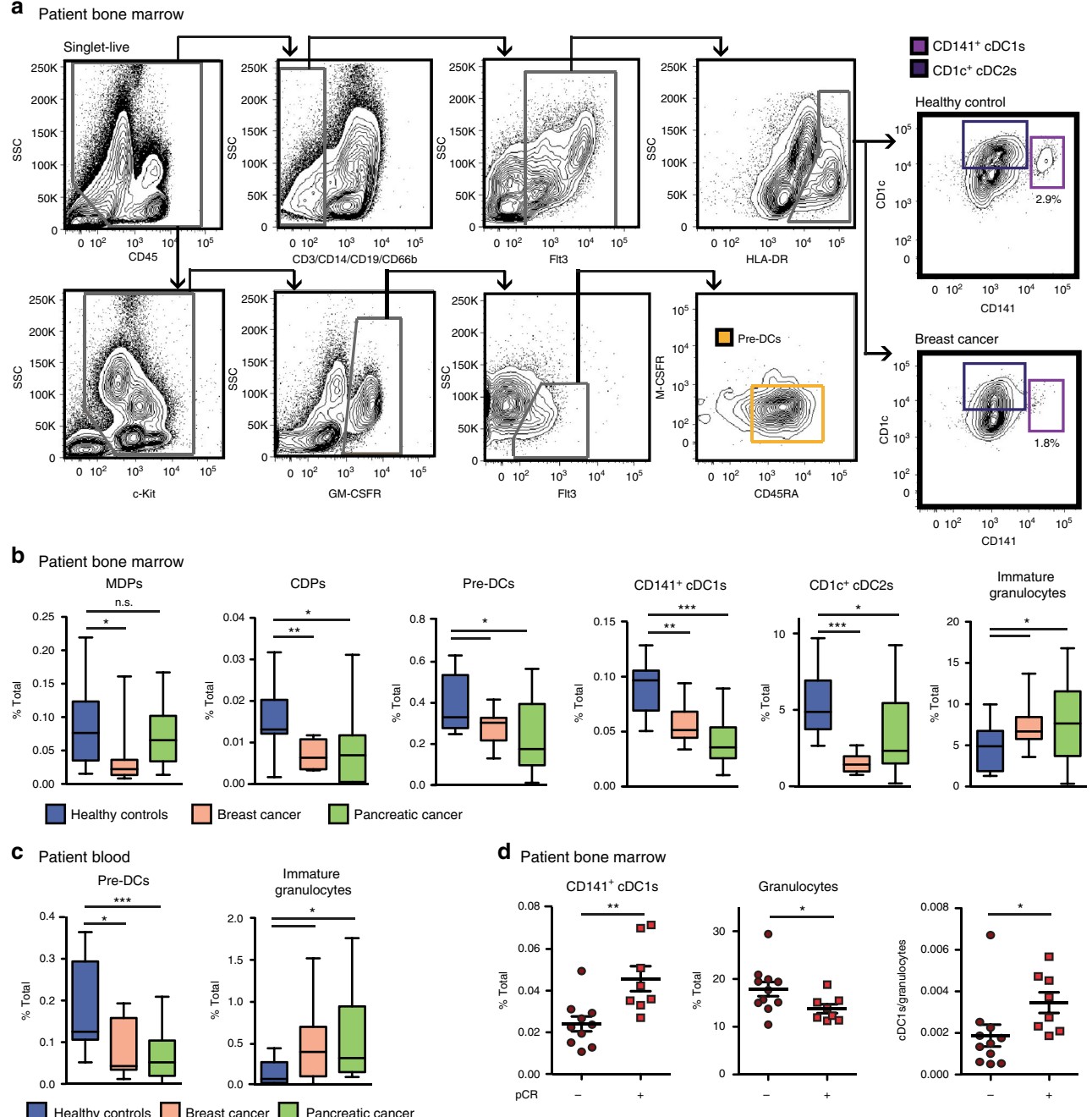

**Fig. 1** Human breast and pancreatic cancers reduce dendritic cell progenitors and CD141$^+$ cDC1s. **a** Representative flow cytometry gating strategy for human BM pre-DCs, CD141$^+$ cDC1s, and CD1c$^+$ cDC2s, including representative final plots from a BC patient and from a healthy control. **b** Frequency of BM MDPs, CDPs, pre-DCs, CD141$^+$ cDC1s, CD1c$^+$ cDC2s, and immature granulocytes (CD11b$^+$CD33$^{High}$CD14$^-$CD15$^+$) in baseline BC and PDAC patients relative to healthy controls. Data from BC cohort 1 and PDAC cohort 1. Healthy controls $n = 12$; BC $n = 10$; PDAC $n = 19$. **c** Frequency of circulating blood pre-DCs and immature granulocytes in baseline BC and PDAC patients relative to healthy controls. Data are from BC cohort 1 and PDAC cohort 1. Healthy controls $n = 12$; BC $n = 10$; PDAC $n = 17$. **d** Frequency of BM CD141$^+$ cDC1s and granulocytes (CD45$^+$CD11b$^+$CSF1R$^-$CD15$^+$CD14$^-$), and the ratio of cDC1s/granulocytes in BC patients prior to treatment or surgical intervention comparing those who achieved pathological complete response relative to patients who did not achieve pathological complete response in BC cohort 2; $n = 18$. Additional cohorts illustrated in Supplementary Figure 1. Error bars represent mean +/− s.e.m. or box plot; *$p < 0.05$, **$p < 0.01$, ***$p < 0.001$, n.s., not significant by unpaired two-sided Student's $t$ test

Fig. 3e, f). Unlike cDC1s, cDC2s were not consistently decreased in the BM of tumor-bearing mice across several models (Fig. 2a–c, Supplementary Fig. 3a–d). Together, these data suggest that solid tumors reduce the number of cells that make up the cDC lineage, including cDC1.

Specific reductions in the cDC progenitors and cDC1s suggest a selective interruption of cDC1 development. Recent data show that pre-DCs can commit to the cDC1 subset before leaving the BM, rather than receiving influence from factors at a peripheral site[27,29,30]. To evaluate this phenomenon, we assessed the number

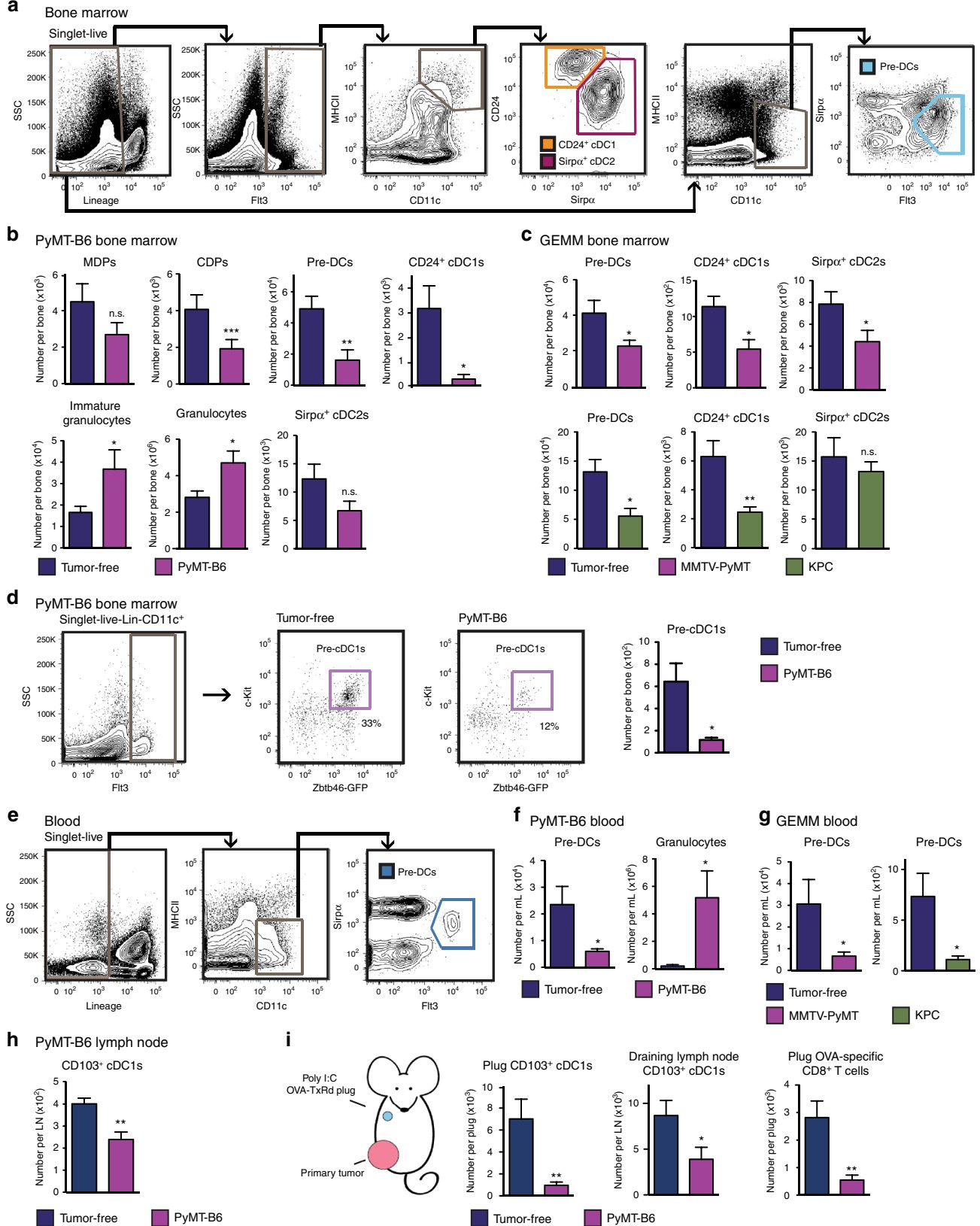

of DC1-committed pre-DCs (pre-cDC1s) in the BM using the gating strategy defined by Grajales-Reyes et al.[29] employing the Zbtb46-GFP mouse. We found that pre-cDC1s were decreased by 80% in the BM of tumor-bearing mice, which is a more substantial decrease than the reduction in the broader pre-DC population (Fig. 2d). These data suggest that tumors not only reduce the number of cDC progenitors, but also reduce the commitment of pre-DCs to the cDC1 lineage in the BM.

To determine if changes in cDC1 progenitors result in systemic alterations in the pool of available cDCs, we analyzed circulating pre-DCs. We found fewer pre-DCs and more granulocytes in the blood of tumor-bearing mice relative to controls (Fig. 2e, f, Supplementary Fig. 2b). Similar observations were seen in the genetic BC model, MMTV-PyMT FVB/N, and genetic PDAC model, KPC (Fig. 2g), as well as two additional orthotopic BC and PDAC models (Supplementary Fig. 3b–c). These changes in the pool of circulating pre-DCs correlated with reduced numbers of CD103$^+$ cDC1s in the uninvolved LNs of tumor-bearing mice (Fig. 2h, Supplementary Fig. 2c). Congruent with observations in BC and PDAC patients, these data show that mouse models of BC and PDAC, independent of genetic driver and strain, have systemically reduced numbers of cDC precursors and cDC1s. Interestingly, pDCs were decreased in the BM, but circulating pDCs were not decreased (Supplementary Fig. 3g), suggesting they may be regulated differently.

Research has shown that cDCs turn over at a high rate in tissues and require BM progenitors to provide new cDC1[17]. To determine if the decreased pool of circulating pre-DCs would result in altered recruitment and antigen presentation at new sites of inflammation, we measured the recruitment of CD103$^+$ cDC1s in the context of a tumor. PyMT-B6 mammary tumors were established in the 4th lower mammary fat pads. Then, a matrigel plug containing polyinosinic:polycytidylic acid (poly I:C) and ovalbumin (OVA)-Texas Red (TxRd) conjugate was implanted in the contralateral 2nd upper mammary fats pads of tumor-bearing and control mice. We found that the presence of the primary tumors resulted in decreased numbers of CD103$^+$ cDC1s recruited into the matrigel plug and decreased numbers of migratory CD103$^+$ cDC1s in the draining LN (Fig. 2i, Supplementary Fig. 2d). These changes in cDC1 recruitment correlated with reduced numbers of OVA-specific CD8$^+$ T cells (CD3$^+$CD8$^+$Dextamer$^+$) in the matrigel plug (Fig. 2i). To verify that the reduced number of cDC1s in the poly I:C plug was due to a decreased pool of circulating pre-DCs rather than changes in recruitment cytokines, we measured recruitment of CD103$^+$

cDC1s to a matrigel plug containing the recruitment factor C-C motif chemokine ligand 4 (Ccl4), which is known to be downregulated in some tumor types[26]. Again, we saw a defect in CD103$^+$ cDC1s recruited to the Ccl4-containing matrigel plug in tumor-bearing mice (Supplementary Fig. 3h). Similarly, we found there was no change in pre-DC C-C motif chemokine receptor 5 (Ccr5) expression in tumor-bearing mice (Supplementary Fig. 3i). Together, these data suggest that tumor-induced repression of cDC development results in a reduced pool of circulating pre-DCs and a subsequent reduction in recruitment of cDC1s to new sites of inflammation and impaired priming of CD8$^+$ T cells.

**Tumors impair cDC1 developmental potential of progenitors.** The contrasting increases in granulocytes and decreases in cDC progenitors led us to hypothesize that tumors alter the differentiation potential of common myeloid precursor cells. To test this, we isolated CD45.1$^+$Lin$^-$cKit$^+$ScaI$^-$ myeloid progenitors (MPs). This cell population maintains granulocyte, monocyte, and cDC potential and contains granulocyte-macrophage progenitors (GMPs)[12]. The MPs were transferred into tumor-free controls or mice bearing PyMT-B6 tumors. We found that CD45.1$^+$ MPs transferred into tumor-bearing mice differentiated into fewer CDPs, pre-DCs, and cDC1s in the BM, but more granulocytes in both the BM and blood (Fig. 3a). Fewer CD8α$^+$ cDC1s were included in the CD45.1$^+$ products in the spleens of tumor-bearing hosts. CD45.1+ granulocyte products did not accumulate in the spleen of tumor-bearing mice relative to tumor-free controls. Although CD45.1$^+$ monocytes were not consistently increased in each organ, they accumulated in the spleen (Supplementary Fig. 4a). These data suggest that tumors preferentially drive the differentiation of shared progenitors to granulocytes at the expense of cDC1 development. To determine if exposure to tumors in vivo programs BM progenitor cells to resist cDC1 differentiation, we isolated MPs, MDPs, or CDPs from tumor-free and PyMT-B6 tumor-bearing mice. When cultured in FMS-like tyrosine kinase 3 ligand (Flt3L), we found MPs, MDPs, and CDPs isolated from tumor-bearing mice were less able to differentiate into cDC1s compared to those cells isolated from tumor-free mice (Fig. 3b). Of these progenitors, CDPs appeared to be most impaired in their capacity to differentiate into cDC1s. Of note, MPs from tumor-bearing mice but not from tumor-free mice differentiated into granulocytes even in the absence of granulocyte differentiation factors (Supplementary Fig. 4b). This result suggests MPs from tumor-bearing mice are

---

**Fig. 2** Primary mammary and pancreatic tumors systemically decrease cDC1s. **a** Representative flow cytometry gating strategy for mouse BM pre-DCs, CD24$^+$ cDC1s, and Sirpα$^+$ cDC2s. **b** Number of BM MDPs, CDPs, pre-DCs, immature granulocytes, granulocytes, CD24$^+$ cDC1s, and Sirpα$^+$ cDC2 from mice bearing end-stage orthotopic PyMT-B6 mammary tumors relative to tumor-free controls; $n = 6$/group. Data are representative of three independent experiments. **c** Number of BM pre-DCs, CD24$^+$ cDC1, and Sirpα$^+$ cDC2 in end-stage genetically engineered mouse models (GEMM) of BC (MMTV-PyMT) and PDAC (KPC) relative to tumor-free controls; tumor-free MMTV-PyMT controls $n = 6$; MMTV-PyMT, $n = 6$; tumor-free KPC controls $n = 7$; KPC, $n = 8$. **d** Alternative representative gating strategy using Zbtb46-GFP mice continued from Fig. 2a for BM committed pre-cDC1s, including representative plots from tumor-free and PyMT-B6 tumor-bearing mice. Number of BM pre-cDC1s in Zbtb46-GFP$^+$ mice bearing end-stage orthotopic PyMT-B6 mammary tumors relative to tumor-free Zbtb46-GFP$^+$ controls; $n = 6$/group. Data are representative of two independent experiments. **e** Representative flow cytometry gating strategy for mouse blood pre-DCs. **f** Number of blood pre-DCs and granulocytes from mice bearing end-stage orthotopic PyMT-B6 mammary tumors relative to tumor-free controls; $n = 6$/group. Data are representative of three independent experiments. **g** Number of blood pre-DCs in end-stage genetic mouse models of BC (MMTV-PyMT) and PDAC (KPC) relative to tumor-free controls, $n = 7$/group. **h** Number of uninvolved LN CD103$^+$ cDC1s from mice bearing end-stage orthotopic PyMT-B6 mammary tumors relative to tumor-free controls; $n = 6$/group. Data are representative of two independent experiments. **i** Mice with 1.0 cm diameter orthotopic PyMT-B6 tumors and tumor-free controls were implanted with matrigel plugs containing poly I:C+OVA-TxRd peptide in the upper mammary fat pad. Number of CD103$^+$ cDC1s in the plug and draining LN and OVA-specific CD8$^+$ T cells (CD3$^+$CD8$^+$Dextamer$^+$) in the plug 10 days after implant; $n = 6$/group. Data are representative of two independent experiments. End-stage for each model is defined in the Methods. Error bars represent mean $+/-$ s.e.m.; $^*p < 0.05$, $^{**}p < 0.01$, $^{***}p < 0.001$, n.s., not significant by unpaired two-sided Student's $t$ test

intrinsically primed to differentiate into granulocytes at the expense of cDC1s. As expected, MDPs and CDPs from tumor-free or tumor-bearing mice did not have granulocyte differentiation potential (Supplementary Fig. 4b, c). Taken together, these data suggest that tumors alter myeloid progenitor fate by increasing their potential to differentiate into granulocytes at the expense of cDC1 production.

To further investigate the differences in progenitors from tumor-free and tumor-bearing mice, we profiled the gene expression in GMPs, MDPs, and CDPs. We found GMPs, MDPs, and CDPs from tumor-bearing mice failed to upregulate proteins indicative of the cDC program (e.g., *Irf8*, *Zbtb46*, *Cd209a*, *Spib*, *Batf3*, and *Bcl11a*). This result is in contrast to the increased expression of markers indicative of a granulocyte or monocyte

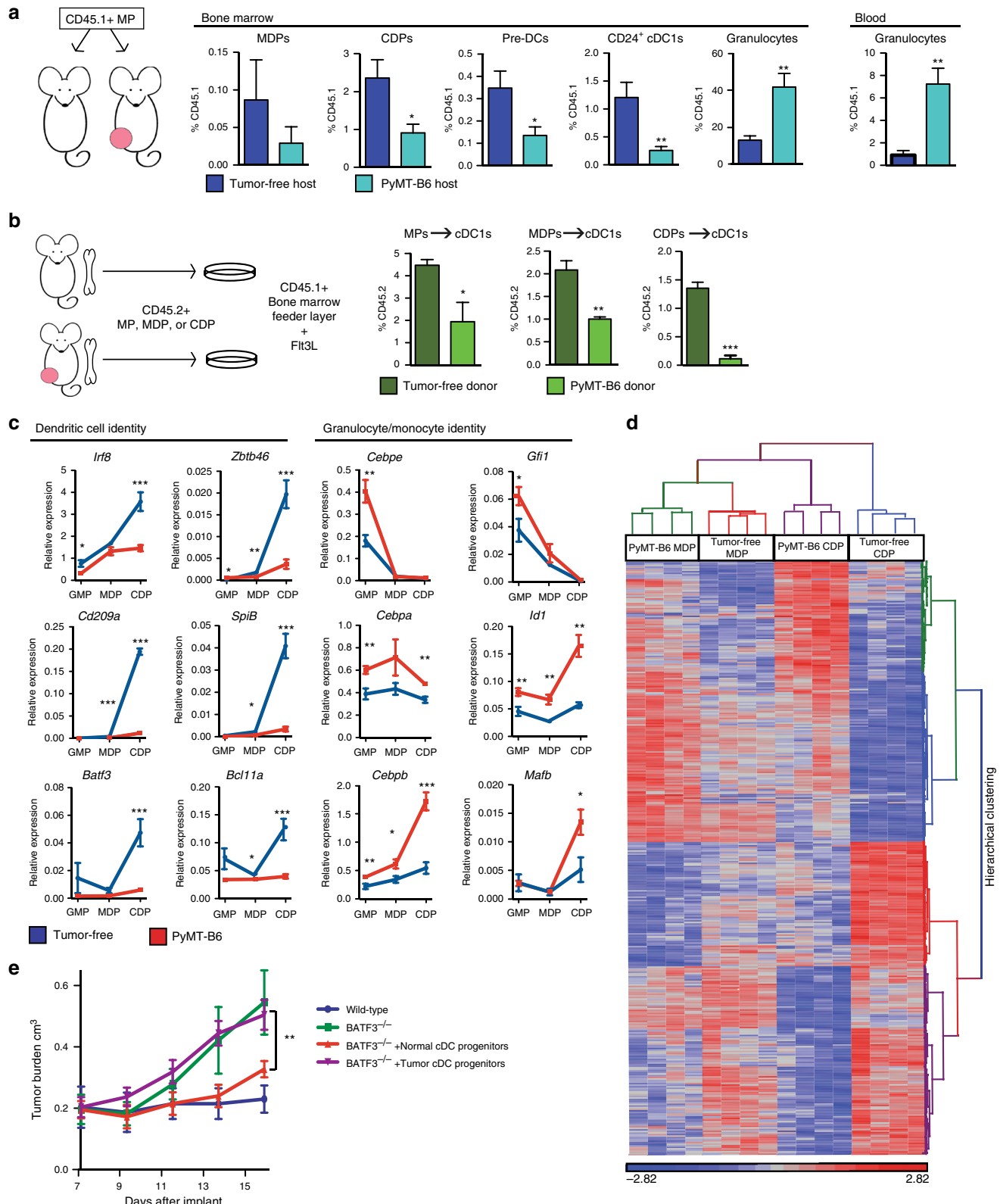

fate (e.g., *Cebpe*, *Gfi1*, *Cebpa*, *Id1*, *Cebpb*, and *Mafb*) (Fig. 3c). We further profiled MDPs and CDPs from tumor-bearing mice by microarray to look for global changes in gene expression. We found the CDPs from tumor-bearing mice clustered closer with MDPs than did CDPs from tumor-free mice (Fig. 3d, Supplementary Tables 2, 3). DAVID (database for annotation, visualization and integrated discovery) analysis showed that the upregulated genes were enriched for biologic process terms like inflammatory response, myeloid cell differentiation, cell cycle regulation, and cytokine secretion, whereas downregulated genes were enriched for biologic process terms like transcriptional regulation, major histocompatibility complex (MHC) class II protein complex binding, differentiation, and covalent chromatin modifications (Supplementary Table 4). Overall, these data suggest a failure in progenitors to upregulate the cDC program, while favoring the granulocytic and monocytic programs in tumor-bearing mice.

To determine if the reduction in cDC1 differentiation potential could impact tumor immunity and tumor progression, we measured the ability of progenitors isolated from tumor-bearing or tumor-free mice to control tumor growth in BATF3$^{-/-}$ mice. We chose BATF3$^{-/-}$ mice because they lack functional cDC1s[18] but have no defect in granulocyte number (Supplementary Fig. 4d). First, we observed that tumor growth of PyMT-mCh-OVA, a variant of PyMT-B6 expressing antigenic mCherry and ovalbumin, was restrained in wild-type, but not in BATF3$^{-/-}$ mice (Fig. 3e). MDPs and CDPs were sorted from PyMT-B6 tumor-bearing or tumor-free controls and transferred into BATF3$^{-/-}$ mice. The recipients were then implanted with PyMT-mCh-OVA. Mice that received MDPs and CDPs from wild-type mice were able to restrain PyMT-mCh-OVA tumor growth, similar to wild-type mice, whereas mice that received MDPs and CDPs from tumor-bearing mice were not able to restrain tumor growth (Fig. 3e). These data suggest that progenitors influenced by tumor burden are no longer able to act as a source of cDC1s and thus fail to induce anti-tumor immunity and control antigenic tumor progression.

**Tumor-derived GCSF inhibits cDC1 development**. To understand which factors inhibit cDC development during cancer progression, we profiled cytokines known to influence myeloid and cDC differentiation in the blood and BM of tumor-bearing mice. We found that both granulocyte colony-stimulating factor (GCSF) and interleukin-6 (IL-6) were highly upregulated in the blood of PyMT-B6 tumor-bearing mice relative to tumor-free controls (Fig. 4a). Additionally, GCSF, but not IL-6, was upregulated in the BM of tumor-bearing mice relative to controls (Fig. 4a) and in the blood serum of BC patients (Fig. 4b). To

determine if either of these cytokines was necessary to alter cDC differentiation, we neutralized both GCSF and IL-6 in the context of PyMT-B6 tumors. As above, the presence of PyMT-B6 tumors reduced BM pre-DC and CD24$^+$ cDC1 and blood pre-DC numbers while increasing granulocytes in both tissues. After treatment with GCSF-neutralizing antibody, we observed that pre-DCs and CD24$^+$ cDC1s in the BM and pre-DCs in the blood were restored to levels in tumor-free mice. This result suggests that tumor-induced GCSF is necessary to reduce cDC1 numbers in tumor-bearing mice. Additionally, GCSF neutralization reduced immature granulocytes in the BM and granulocytes in the blood, as expected[32,37]. Neutralization of GCSF was also sufficient to increase CD8$^+$ T cells in the tumor (Fig. 4c). Although IL-6 neutralization reduced immature granulocytes and inflammatory monocytes as previously described[38], it did not reverse tumor-induced reductions in pre-DCs or CD24$^+$ cDC1s in the BM (Supplementary Fig. 5a).

To determine if GCSF alone is sufficient to disrupt cDC differentiation, we dosed mice with recombinant GCSF. We used 2 μg/day of recombinant GCSF to achieve steady-state serum GCSF concentrations comparable to those in tumor-bearing mice (Supplementary Fig. 5b). We found that changes in BM progenitor numbers in tumor-free mice dosed with GCSF mice paralleled those observed in tumor-bearing mice. These changes included GCSF-induced reduction in MDPs, CDPs, pre-DCs, and cDC1s in the BM and reduced pre-DCs in the blood as compared to untreated mice (Fig. 4d). As expected, GCSF increased immature granulocytes in the BM and granulocytes in the blood (Fig. 4d). These data implicate a role for GCSF in reducing cDC1 numbers during cancer progression.

To determine if GCSF could act directly on cDC progenitors to inhibit cDC1 differentiation, we cultured MPs, MDPs, and CDPs isolated from tumor-free mice in the presence of Flt3L+/- GCSF. As expected, Flt3L drove cDC1 differentiation in all cell types, whereas GCSF prevented Flt3L-mediated cDC1 expansion, and in MPs instead drove granulocyte differentiation (Fig. 4e). These results are consistent across a wide range of Flt3L and GCSF concentrations (Supplementary Fig. 5c). To understand if GCSF alone could prime progenitors to become unresponsive to Flt3L-mediated cDC differentiation, consistent with the ex vivo results shown in Fig. 3b, progenitors were pre-treated with GCSF for 24 h prior to plating for differentiation in Flt3L. Pre-treatment with GCSF inhibited cDC1 differentiation even in CDPs (Fig. 4f), suggesting that GCSF can alter the ability of a progenitor to respond to cDC differentiation cues even in progenitors lacking granulocyte differentiation capacity. To ensure GCSF was able to signal in cDC progenitors directly, we analyzed the expression of the GCSF receptor, *Csf3r*, in vivo. We found that *Csf3r* was expressed in GMPs, MDP, and CDPs and unmodified by the

---

**Fig. 3** Tumor burden alters the fate of myeloid progenitors. **a** CD45.1$^+$Lin$^-$Sca1$^-$cKit$^+$ MPs were transferred into mice bearing 1.0 cm diameter orthotopic PyMT-B6 tumors or tumor-free controls. BM was analyzed for CD45.1$^+$ populations after 2 weeks. BM MDPs, CDPs, pre-DCs, CD24$^+$ cDC1s, and granulocytes, and blood granulocytes displayed as frequency of CD45.1; $n = 5$/group. **b** CD45.2$^+$Lin$^-$Sca1$^-$cKit$^+$ MPs, CD45.2$^+$ MDPs, and CD45.2$^+$ CDPs were isolated from end-stage orthotopic PyMT-B6 tumor-bearing or tumor-free donors. Progenitors were cultured on CD45.1$^+$ BM feeder culture for 5 days in the presence of 100 ng/ml Flt3L. Final cultures were analyzed for cDC1 (Live CD45.2$^+$CD45.1$^-$MHCII$^+$CD11c$^+$Sirpα$^-$CD24$^+$). End stage is defined in the Methods. Data are representative of three independent experiments consisting of three wells per condition. GMPs, MDPs, and CDPs were sorted from mice bearing end-stage orthotopic PyMT-B6 mammary tumors and tumor-free controls. **c** GMPs, MDPs, and CDPs were analyzed by RT-qPCR. **d** MDPs and CDPs were analyzed by microarray. Cluster analysis was performed with a differential genes list generated from gene with 1.5-fold at $p < 0.05$ and FDR $q < 0.05$ in MDP or CDP comparison from PyMT-B6 tumor bearing to tumor free. Four samples, each consisting of two mice, were analyzed per group. **e** MDPs and CDPs were isolated from PyMT-B6 tumor-bearing mice or tumor-free controls. Progenitors were adoptively transferred into BATF3$^{-/-}$ mice. PyMT-mCh-OVA was implanted after 3 days into wild-type mice, BATF3$^{-/-}$ mice without adoptive transfer, and BATF3$^{-/-}$ mice with adoptive transfer from tumor-free or tumor-bearing mice. Tumor growth was monitored. Error bars represent mean +/− s.e.m.; *$p < 0.05$, **$p < 0.01$, ***$p < 0.001$ by unpaired two-sided Student's $t$ test or two-way ANOVA

presence of tumors (Supplementary Fig. 5d). Together, these data suggest that tumor-induced GCSF is both necessary and sufficient to reduce cDC1 differentiation and can act directly on cDC progenitors.

We next sought to determine the source of GCSF. First we analyzed both tumor tissues and BM samples from the transplantable PyMT-B6 and the genetic KPC models for *Csf3* by in situ hybridization (ISH). We found that although BM cells expressed *Csf3*, levels were not elevated by the presence of PyMT-B6 and KPC tumors (Supplementary Fig. 5e). By contrast, tumor tissues from PyMT-B6 and KPC models had higher *Csf3* levels

compared to normal mammary and pancreas tissues (Fig. 5a). We also observed that the majority of the *Csf3* ISH signal was present in the tumor cells and not the stroma (Fig. 5a). To verify this result in human BC and PDAC tissues, we stained for GCSF protein by immunohistochemistry (IHC). Similar to our mouse models, we found that GCSF was highly expressed in breast and pancreas tumor tissues (Fig. 5b). Additionally, although GCSF was expressed in both tumor and stroma, tumor levels of GCSF were considerably higher than stroma levels in most cases. Together, these data suggest that tumor-derived GCSF may be a dominant source of GCSF in tumor-bearing animals. To directly

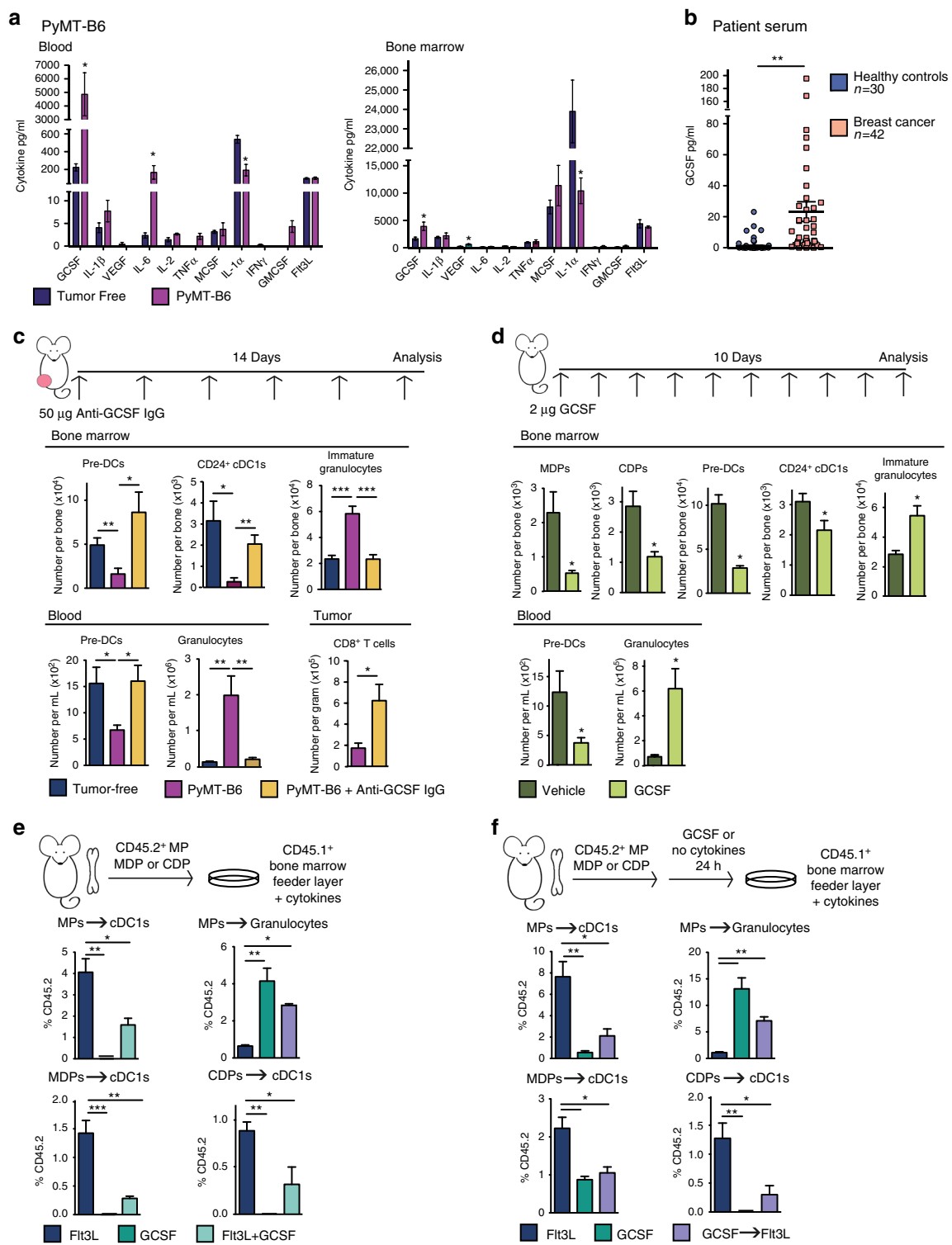

test if tumor-derived GCSF was responsible for impaired cDC development, we deleted the *Csf3* gene using CRISPR CAS9 in the PyMT-B6 cell line (PyMT-B6 GCSFKO). Both BM and blood profiling showed pre-DCs and cDC1 populations were recovered and immature granulocytes were reduced in PyMT-B6 GCSFKO tumor-bearing mice relative to PyMT-B6 tumor-bearing controls (Fig. 5c). These results were repeated with a second independent clone (Supplementary Fig. 5f). These data demonstrate that GCSF derived from tumor cells is impairing cDC1 differentiation.

**GCSF inhibits cDC1 development by impairing IRF8 expression.** Research has shown that GCSF preferentially expands immature granulocytes and monocytes by suppressing IRF8 expression through activation of signal transducer and activator of transcription 3 (STAT3) in granulocytic-monocytic precursors[39,40]. Because IRF8 is an important transcription factor for cDC1 development, we hypothesized GCSF might drive reduced IRF8 expression in cDC progenitors leading to impaired cDC1 differentiation[19,41,42]. In support of this notion, our expression profiling demonstrated that *Irf8* messenger RNA (mRNA) was downregulated in cDC progenitors from tumor-bearing mice (Fig. 3c). Additionally, we found IRF8 protein was reduced in tumor-bearing mouse MDPs, CDPs, and pre-DCs (Fig. 6a). Further, we analyzed IRF8 target gene expression in GMPs, MDPs, and CDPs and found that, relative to controls, tumor-bearing mice had decreased mRNA expression of multiple MHC molecules, as well as *TapBP*, *Tap2*, *Batf3*, and *Pml* (Figs. 3c, 6b). In contrast, *Id2*, another regulator of cDC1 development, was not reduced (Supplementary Fig. 6a)[17]. To understand if down-regulation of IRF8 is responsible for some of the gene expression changes in the microarray analysis (Fig. 3d), differentially expressed genes between control and tumor-bearing mice were compared to genes that are differentially expressed in IRF8$^{-/-}$ myeloid progenitors[39,41]. We found tumor-regulated genes were enriched for genes regulated in IRF8$^{-/-}$ myeloid progenitors (Supplementary Fig. 6b). In keeping with the ability of GCSF to downregulate IRF8 through STAT3 activation[39], we observed that phosphorylated STAT3 (pSTAT3) was elevated in MDPs, CDPs, and pre-DCs from tumor-bearing mice (Fig. 6c).

To determine if these changes also occur in human cancer patients, we analyzed IRF8 expression in human BM and found that IRF8 was downregulated in both BC and PDAC patients compared to healthy controls (Fig. 6d). Additionally, the extent of IRF8 downregulation correlated with the decrease in BM pre-DCs in patients (Fig. 6e). We also found IRF8 expression was decreased in the blood pre-DCs of pancreatic cancer patients (Fig. 6f) and the extent of IRF8 downregulation correlated with blood pre-DC numbers (Fig. 6g). Because IRF8 is a molecular marker of pre-DCs committed to the cDC1 lineage, these data

also suggest that patients have reduced numbers of circulating pre-cDC1s relative to healthy controls. We next assessed if downregulation of IRF8 in circulating pre-DCs impacted patient outcome and found patients with low IRF8 expression in circulating pre-DCs had decreased recurrence-free and overall survival (Fig. 6h). These results suggest that IRF8 expression is modulated in human cancers and may be an indicator of cDC1 development and patient outcome.

To confirm that GCSF was upstream of IRF8 suppression, we measured IRF8 expression in cDC progenitors from tumor-bearing mice following GCSF neutralization. We found IRF8 expression was recovered in the MDP, CDP, and pre-DCs following anti-GCSF immunoglobulin G (IgG) treatment (Fig. 7a). Furthermore, IRF8 expression was, again, downregulated in progenitors from PyMT-B6 tumor-bearing mice but was not downregulated in progenitors from PyMT-B6 GCSFKO mice (Supplementary Fig. 6c). To show GCSF was acting directly on progenitors to modulate IRF8 expression, we treated MPs, MDPs, and CDPs alone in vitro with GCSF or GCSF+Flt3L. We found *Irf8*, as well as its target gene *Batf3*, were downregulated in both conditions relative to controls (Fig. 7b). Interestingly, Flt3L treatment did not drive IRF8 expression above controls (Supplementary Fig. 6d), suggesting Flt3L and GCSF act through different mechanisms to modulate cDC1 differentiation. To determine if downregulation of IRF8 is necessary for GCSF to impair cDC1 differentiation, we investigated if IRF8 overexpression would make MPs insensitive to GCSF. Without IRF8 overexpression, GCSF drove down Flt3L-mediated cDC1 differentiation, similar to the results shown in Fig. 4e. However, IRF8 overexpression rendered MPs insensitive to GCSF, and their cDC1 differentiation was comparable to Flt3L-alone conditions (Fig. 7c). These data suggest GCSF must modulate IRF8 expression to impact cDC1 differentiation.

**IRF8 and cDC1s support anti-tumor CD8$^+$ T-cell responses.** To determine whether suppression of cDC1 differentiation was detrimental to anti-tumor immunity, we measured tumor-specific T-cell activation in the presence or absence of an established tumor. To accomplish this, PyMT-B6 tumors, referred to here as the primary tumor, were established in the lower mammary fat pad. A secondary tumor using the PyMT-mCh-OVA cell line was then implanted into the upper mammary fat pad of the same mouse. We observed reduced recruitment of CD103$^+$ cDC1s to the secondary tumor site and draining LNs of primary tumor-bearing mice (Fig. 8a). This result is consistent with the reduction in the available pool of circulating pre-cDC1s in tumor-bearing animals (Fig. 2i). To test if reduced recruitment of CD103$^+$ cDC1s was functionally relevant, we measured OVA-specific CD8$^+$ T cells within the secondary tumor site and the secondary tumor-draining LNs and found the OVA-specific CD8$^+$ T cells

**Fig. 4** GCSF disrupts cDC1 differentiation. **a** Blood and BM serum cytokines in orthotopic PyMT-B6 end-stage tumor-bearing mice relative to tumor-free controls; tumor-free, $n = 6$; PyMT-B6, $n = 7$. **b** GCSF in human BC patient blood serum relative to healthy controls; healthy controls, $n = 30$; BC, $n = 42$. **c** Number of BM pre-DCs, CD24$^+$ cDC1s, and immature granulocytes; numbers of blood pre-DCs and granulocytes, and tumor CD8$^+$ T cells (CD45$^+$CD3$^+$CD8$^+$) in tumor-free mice, orthotopic PyMT-B6 tumor-bearing end-stage mice, and orthotopic PyMT-B6 tumor-bearing end-stage mice treated for 2 weeks with 50 μg anti-GCSF IgGs 3 ×/week; $n = 5$-7/group. Data are representative of two independent experiments. **d** Number of BM MDPs, CDPs, pre-DCs, CD24$^+$ cDC1s, and immature granulocytes, and number of blood pre-DCs and granulocytes in C57BL/6 mice treated with 2 μg GCSF for 10 days; $n = 6$/group. Data are representative of two independent experiments. **e** CD45.2$^+$Lin$^-$Sca1$^-$cKit$^+$ MPs, MDPs, and CDPs isolated from tumor-free mice were cultured on CD45.1$^+$ BM feeder culture in the presence of 100 ng/ml Flt3L, 100 ng/ml GCSF, or 100 ng/ml Flt3L and 100 ng/ml GCSF for 5 days. Final cultures were analyzed for cDC1s (Live CD45.2$^+$CD45.1$^-$MHCII$^+$CD11c$^+$Sirpα$^-$CD24$^+$) and granulocytes (Live CD45.2$^+$CD45.1$^-$CD11b$^+$Ly6G$^+$). Data are representative of three independent experiments consisting of three wells per condition. **f** Experiment similar to that in Fig. 4d, but CD45.2$^+$ cells were pre-treated with 100 ng/ml GCSF or media alone for 24 h prior to plating on CD45.1$^+$ BM feeder layer. Data are representative of two independent experiments consisting of three wells per condition. End stage for each model is defined in the Methods. Error bars represent mean+/− s.e.m. or box plot; *$p < 0.05$, **$p < 0.01$, ***$p < 0.001$ by unpaired two-sided Student's $t$ test

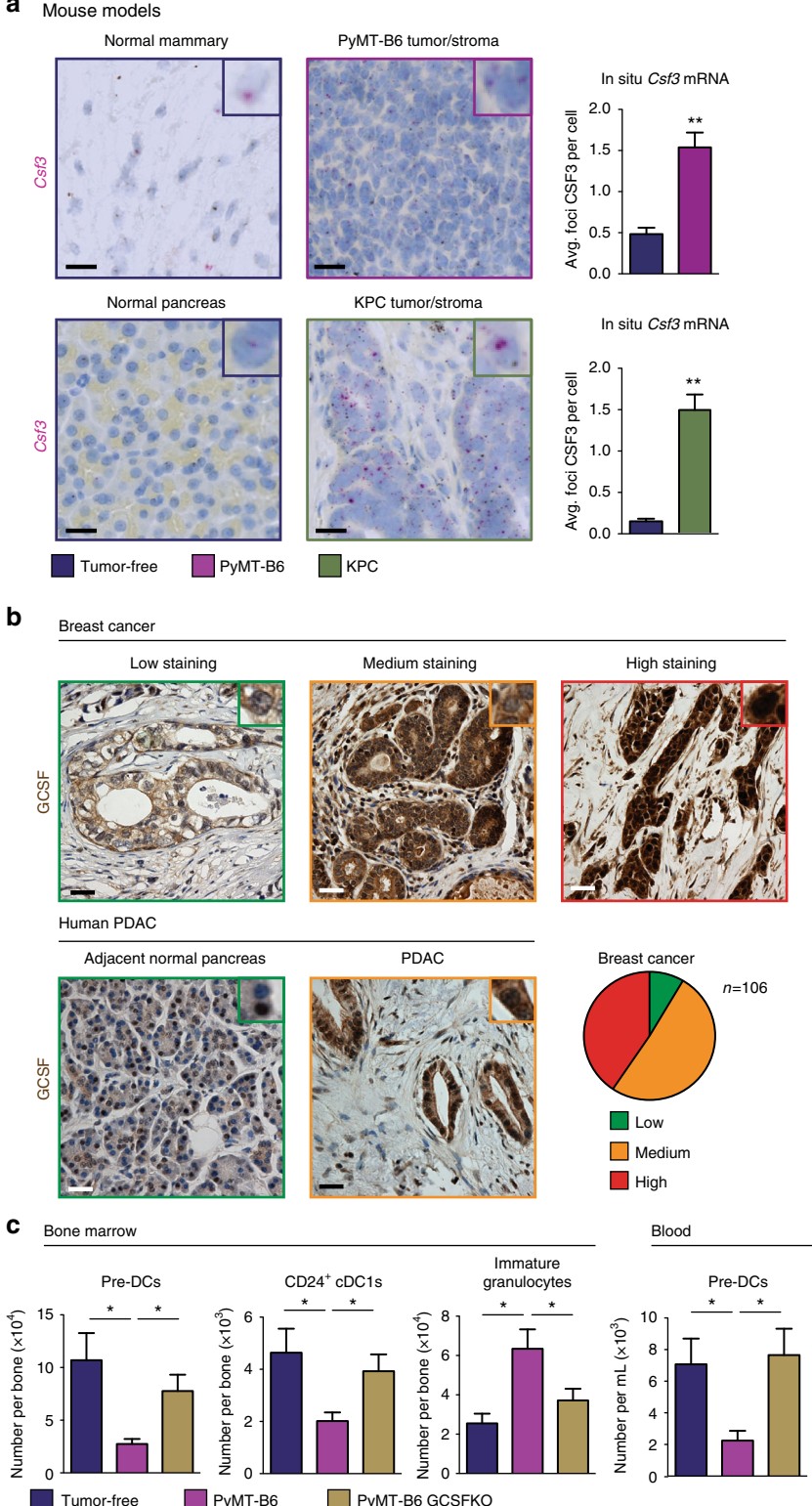

**Fig. 5** Tumor-derived GCSF inhibits cDC1 development. **a** ISH for *Csf3* on end-stage PyMT-B6 and KPC tumor tissue and tumor-free mammary or pancreas tissue; $n = 3$ per group. Scale bar 25 μm. **b** Human patient BC and PDAC tumor tissue stained for GCSF. BC graded for low, medium, and high staining per tumor cell; BC, $n = 106$; PDAC, $n = 5$. Scale bar 10 μm. **c** Number of BM pre-DCs, CD24[+] cDC1s, and immature granulocytes and blood pre-DCs from mice bearing end-stage orthotopic PyMT-B6 or PyMT-B6 GCSFKO mammary tumors relative to tumor-free controls, $n = 6$/group. End stage for each model is defined in the Methods. Error bars represent mean +/− s.e.m.; *$p < 0.05$, **$p < 0.01$ by unpaired two-sided Student's *t* test

were reduced in both sites of primary tumor-bearing mice compared to controls (Fig. 8a). To specifically implicate IRF8 and cDC1s in this decreased CD8$^+$ T-cell response, we employed IRF8$^{-/-}$ and Batf3$^{-/-}$ mice, both of which have impaired cDC1 differentiation[18,19,42]. In these knockouts, we observed reduced numbers of tumor-specific CD8$^+$ T cells comparable to the primary tumor-bearing mice. In addition, this reduction was not further decreased by the presence of primary tumors (Fig. 8a). These data suggest that tumor-induced reductions in cDC1 development have a functional consequence on tumor-specific CD8$^+$ T-cell responses that could contribute to reduction in anti-tumor immunity.

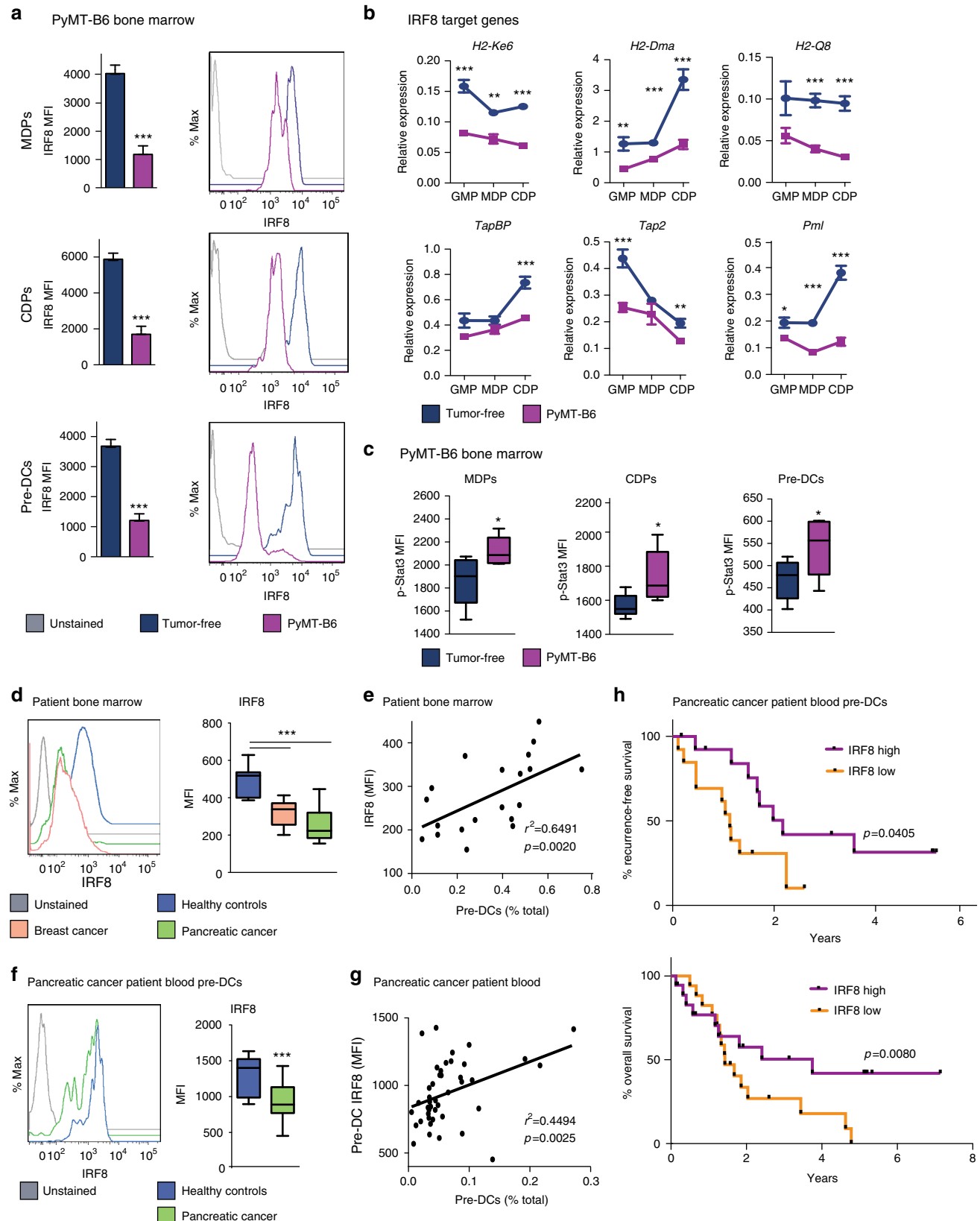

Tumor-induced reductions in IRF8 also expand immature granulocytes that are known to have immune-suppressive functions and can otherwise promote tumor progression[8,37,43]. We sought to determine the contributions of immature granulocyte expansion to reduced number of cDC1s and reduced anti-tumor CD8+ T-cell responses. To test if cDC1 depletion was regulated through granulocyte expansion, mice were treated with anti-Ly6G IgGs. Ly6G-depletion did not reverse the defect in BM pre-DCs and cDC1s and did not recover IRF8 expression in BM progenitors (Supplementary Fig. 7a). To show that immature granulocytes were not mediating CD8+ T-cell suppression beyond the loss of cDC1s in this model, we neutralized Ly6G in the context of the secondary tumor experiment described above. We found Ly6G depletion did not recover tumor-specific CD8+ T cells in the secondary tumor nor the draining LN (Fig. 8a). Similarly, we showed anti-Ly6G depletions did not recover CD103+ cDC1 recruitment to a matrigel plug containing poly I:C and OVA-TxRd conjugate, a secondary site of inflammation, in primary tumor-bearing mice. Additionally, CD103+ cDC1s were not recovered in the draining LN and OVA-specific CD8+ T cells were not recovered in the matrigel plug (Supplementary Fig. 7b). To test whether the loss of IRF8 had a functional consequence on tumor outgrowth by acting through the loss of cDC1s rather than the expansion of granulocytes, we measured the growth of PyMT-mCh-OVA tumors in IRF8−/− mice, which lack cDC1s and have expanded granulocytes, and Batf3−/− mice, which only lack functional cDC1 (Supplementary Fig. 7c)[18,41]. We found that tumors grew at the same rate in Batf3−/− as IRF8−/− mice, but that this rate was faster than wild-type controls (Fig. 8b). Also, we neutralized Ly6G in IRF8−/− mice and measured PyMT-mCh-OVA tumor growth and found tumors grew at similar rates (Fig. 8c). These findings suggest that the loss of IRF8 can promote tumor growth by suppressing cDC1s rather than solely through the expansion of granulocytes. Together, these results show granulocytes do not act as an intermediate to inhibit cDC1 differentiation and the functional effects on antigen specific CD8+ T cells and tumor control are primarily mediated by cDC1s, not granulocytes, in this model.

**GCSF neutralization overcomes resistance to immunotherapy.** Our data suggest that exposure to tumor-derived GCSF can impair BM progenitors in their ability to generate cDC1s. As Flt3L therapy has been developed for use in patients to bolster cDC numbers and function[2], we asked if the impairment in cDC development could be rescued by Flt3L treatment in vivo or if GCSF neutralization was necessary for maximum efficacy of this therapy. To accomplish this we treated tumor-free or mice bearing a 1 cm+ diameter PyMT-B6 tumor with Flt3L+/− anti-GCSF IgGs for 2 weeks. Consistent with previous reports[2], we

observed that in tumor-free mice treated with Flt3L expanded pre-DCs, cDC1, and cDC2 in the BM and pre-DCs in the blood (Fig. 9a, b). We also observed that combined treatment with Flt3L and anti-GCSF IgGs did not further upregulate pre-DC or cDC numbers in tumor-free mice (Fig. 9a, b). This result is consistent with the finding that GCSF is expressed at lower levels under non-pathologic conditions (Fig. 4a). Similar to our other experiments, pre-DC and cDC1 numbers were decreased in the BM and blood of PyMT-B6 tumor-bearing mice compared to controls (Fig. 9a, b). We also found that although Flt3L could expand pre-DC and cDC numbers in the BM and blood in the presence of tumors, this expansion was limited compared to tumor-free mice in that pre-DC or cDC1 numbers only reached levels comparable to untreated tumor-free mice (Fig. 9a, b). These data suggest that BM progenitors in tumor-bearing mice are resistant to Flt3L-induced cDC1 differentiation, which is consistent with our in vitro results (Fig. 3b). Furthermore, anti-GCSF IgGs and Flt3L synergized to recover pre-DC and cDC1 numbers in tumor-bearing mice (Fig. 9a, b). Because our data suggest that GCSF impacts IRF8 levels, we analyzed the impact of Flt3L and anti-GCSF IgGs on IRF8 levels in DC progenitors. In tumor-bearing mice, in which IRF8 is downregulated, neutralizing GCSF restores IRF8 levels to tumor-free mouse conditions in BM progenitors. Importantly, neutralizing GCSF also increased IRF8 levels in pre-DCs (Fig. 9c, Supplementary Fig. 7d). These findings suggest neutralizing tumor-induced GCSF is required to recover IRF8 expression, which is necessary for cDC1 differentiation.

To determine if these changes in lymphoid tissues translated to increased tumor immunity, we analyzed tumor tissues. As previously reported[2], Flt3L increases both tumor-infiltrating CD103+ cDC1s and CD11b+ cDC2s, but this expansion of cDC1s was modest in this model (Fig. 9d). When Flt3L was combined with anti-GCSF IgGs, tumor infiltration of CD103+ cDC1s, but not CD11b+ cDC2s, were increased by more than twofold and this increase correlated with substantially more CD8+ T cells. Together, these data show that GCSF neutralization supports Flt3L-mediated expansion and mobilization of cDC1 and suggest this activity is facilitated in part by restoring IRF8 expression and commitment to the cDC1 lineage.

Given these results, we hypothesized Flt3L+anti-GCSF IgGs would bolster cDC-based immunotherapeutic strategies in established tumors. We employed poly I:C, which supports cDC maturation, and anti-programmed cell death protein 1 (PD1) IgGs, which block a T-cell checkpoint. Mice bearing well-established 1 cm+ diameter PyMT-B6 tumors were treated with Flt3L+poly I:C+anti-PD1 IgGs+/−anti-GCSF IgGs. Mice treated with combined Flt3L+poly I:C+anti-PD1 IgGs did not have an increase in survival relative to vehicle-treated controls. By contrast, survival was extended in mice treated with anti-GCSF IgGs+Flt3L+poly I:C+anti-PD1 IgGs compared with mice treated with vehicle alone or Flt3L+poly I:C+anti-PD1 IgGs

**Fig. 6** IRF8 expression is reduced during breast and pancreatic cancer. **a** IRF8 measured in BM MDPs, CDPs, and pre-DCs of end-stage orthotopic PyMT-B6 tumor-bearing mice and tumor-free controls; $n = 7$/group. Data are representative of three independent experiments. **b** RT-qPCR analysis of BM GMPs, MDPs, and CDPs sorted from mice bearing end-stage orthotopic PyMT-B6 mammary tumors and tumor-free controls for IRF8 target genes, four samples consisting of two mice each were analyzed per group. **c** pSTAT3 measured in BM MDPs, CDPs, and pre-DCs of end-stage orthotopic PyMT-B6 tumor-bearing mice and tumor-free controls; $n = 6$. Data are representative of two independent experiments. **d** IRF8 expression in total BM from BC and PDAC patients relative to healthy controls; $n = 10$/group. Data are from BC cohort 1 and PDAC cohort 1. **e** Correlation between BM pre-DCs and BM IRF8 expression in BC and PDAC patients. Data are from BC cohort 1 and PDAC cohort 1; $n = 20$. **f** IRF8 expression in blood pre-DCs from PDAC patients relative to healthy controls; healthy controls $n = 10$, PDAC $n = 43$. Data are from PDAC cohort 2. **g** Correlation between blood pre-DCs and blood pre-DC IRF8 expression in PDAC patients; $n = 43$. Data are from PDAC cohort 2. **h** Kaplan–Meier estimate of recurrence free survival and overall survival of patients +/− median blood pre-DCs IRF8 expression. Log-rank (Mantel–Cox) $p$ value is denoted for differences in recurrence-free survival and overall survival. Data are from PDAC cohort 2. Spearman's correlation ($r^2$, correlation coefficient). End stage for each model is defined in the Methods. Error bars represent mean +/− s.e.m or box plot; *$p < 0.05$, **$p < 0.01$, ***$p < 0.001$ by unpaired two-sided Student's $t$ test

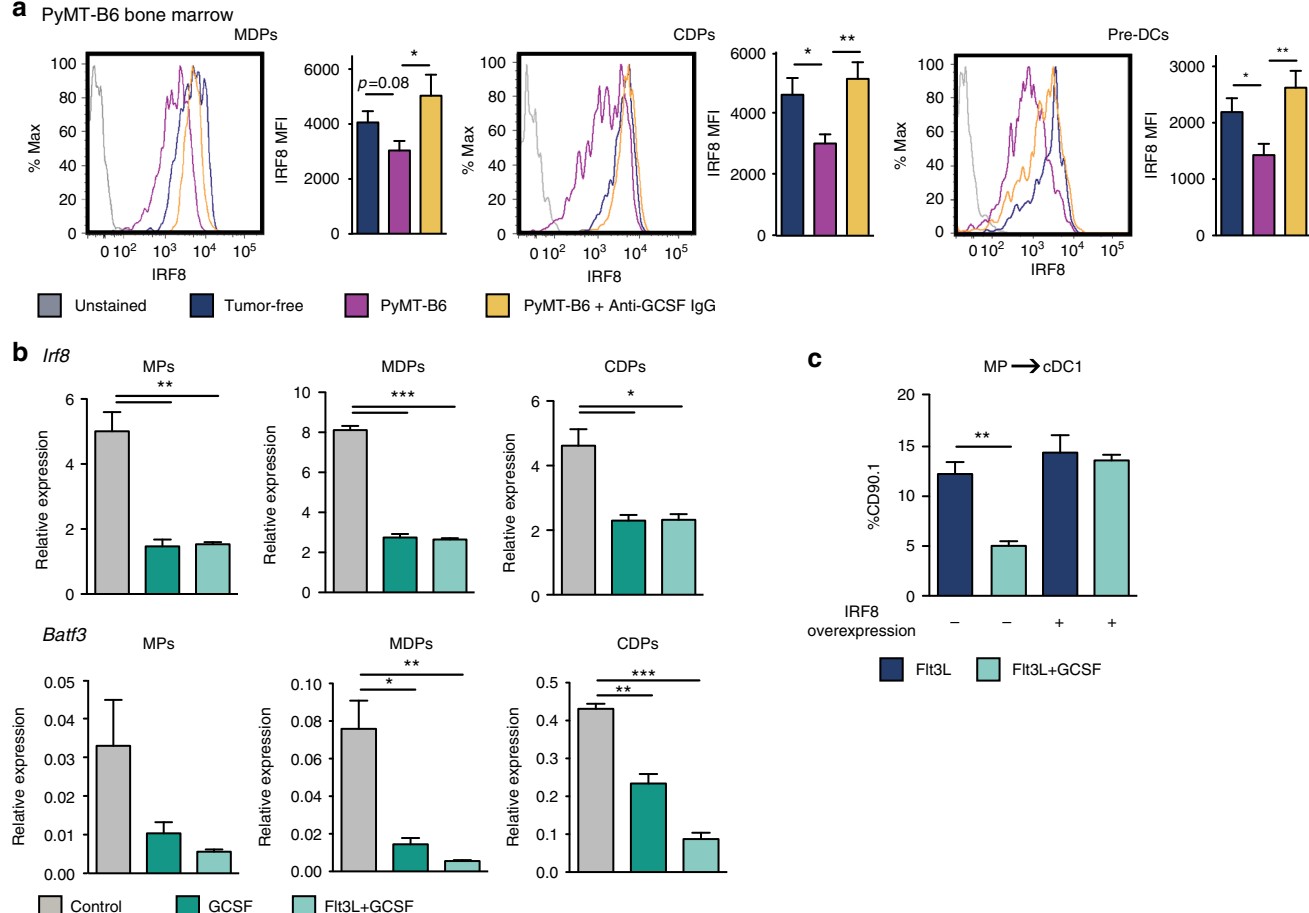

**Fig. 7** GCSF regulates IRF8 in dendritic cell progenitors. **a** IRF8 measured in BM MDPs, CDPs, and pre-DCs of tumor-free mice, orthotopic PyMT-B6 tumor-bearing end-stage mice, and orthotopic PyMT-B6 tumor-bearing end-stage mice treated for 2 weeks with 50 μg anti-GCSF IgGs 3 × /week; $n = 6$/group. Data are representative of two independent experiments. End stage is defined in the Methods. **b** MPs, MDPs, and CDPs treated with 100 ng/ml GCSF and/or 100 ng/ml Flt3L for 24 h. Analyzed by RT-qPCR for *Irf8* and *Batf3*. Data are representative of two independent experiments consisting of three wells per condition. **c** CD45.2+Lin−Sca1−cKit+ MPs transduced with CD90.1 IRF8 overexpression vector or CD90.1 empty vector control were cultured on CD45.1+ BM feeder culture in the presence of 100 ng/ml Flt3L and/or 100 ng/ml GCSF for 3 days. Final cultures were analyzed for cDC1s (Live CD45.2+CD90.1+CD45.1−MHCII+CD11c+Sirpα−CD24+). Data are representative of three independent experiments consisting of three wells per condition. Error bars represent mean +/− s.e.m. or box plot; *$p < 0.05$, **$p < 0.01$, ***$p < 0.001$ by unpaired two-sided Student's *t* test

(Fig. 9e, Supplementary Fig. 7e). This effect was not seen in similarly treated IRF8−/− and BATF3−/− mice, suggesting that this treatment is reliant on IRF8 and cDC1 activity (Fig. 9e, Supplementary Fig. 7e). Together, these data suggest that GCSF signaling may need to be inhibited to employ immunotherapy effectively in some established tumor settings.

## Discussion

Breast and pancreas cancers have had limited responses to single agent immunotherapy in the clinic[44–47]. Although this limited response might be explained in part by poor antigenicity, BC and PDAC are also known to expand systemic populations of immune-suppressive myeloid cells by increasing differentiation from myeloid progenitors in the BM[7,8,44]. Our findings suggest that tumor-induced inflammatory cytokines expand potentially immune-suppressive myeloid cells and simultaneously suppress anti-tumor cDC1 development from BM progenitors. Beyond committing to the granulocyte, monocyte, or cDC lineages, cDC progenitors further commit to the cDC1 subset before leaving the BM[27–30]. These findings suggest that the antigen-presenting capacity can be defined before cells traffic into the periphery and

is not entirely reliant on cues experienced at the tumor site. Therefore, it is important to understand how tumors alter BM myeloid differentiation to overcome tumor-induced immune surveillance. Others have shown suppression of the transcription factor IRF8 during tumor progression expands the immature granulocyte and monocyte populations that are known to suppress anti-tumor immune responses[39–41]. We further demonstrated IRF8 downregulation during cancer reduces cDC1 development in the BM. This is an important finding because IRF8 expression in progenitors as early as the hematopoietic stem cell primes transcriptional networks and influences lineage bias towards cDC and cDC1 fate[48,49]. Additionally, tumor-induced depletion of cDC1s has a functional implication on anti-tumor immunity beyond the effect of granulocyte expansion. Even in the context of granulocyte depletion, reduced cDC1 differentiation inhibits CD8+ T cell-mediated anti-tumor responses, leading to a loss of tumor control in this model. Given that previous work has shown granulocyte depletion is sufficient to recover anti-tumor immunity in some cases[9,50–53], we speculate there is a balance between the expansion of granulocytes as immune suppressors and depletion of cDC1s as immune stimulators that varies with tumor type, disease progression, and tumor models. In addition

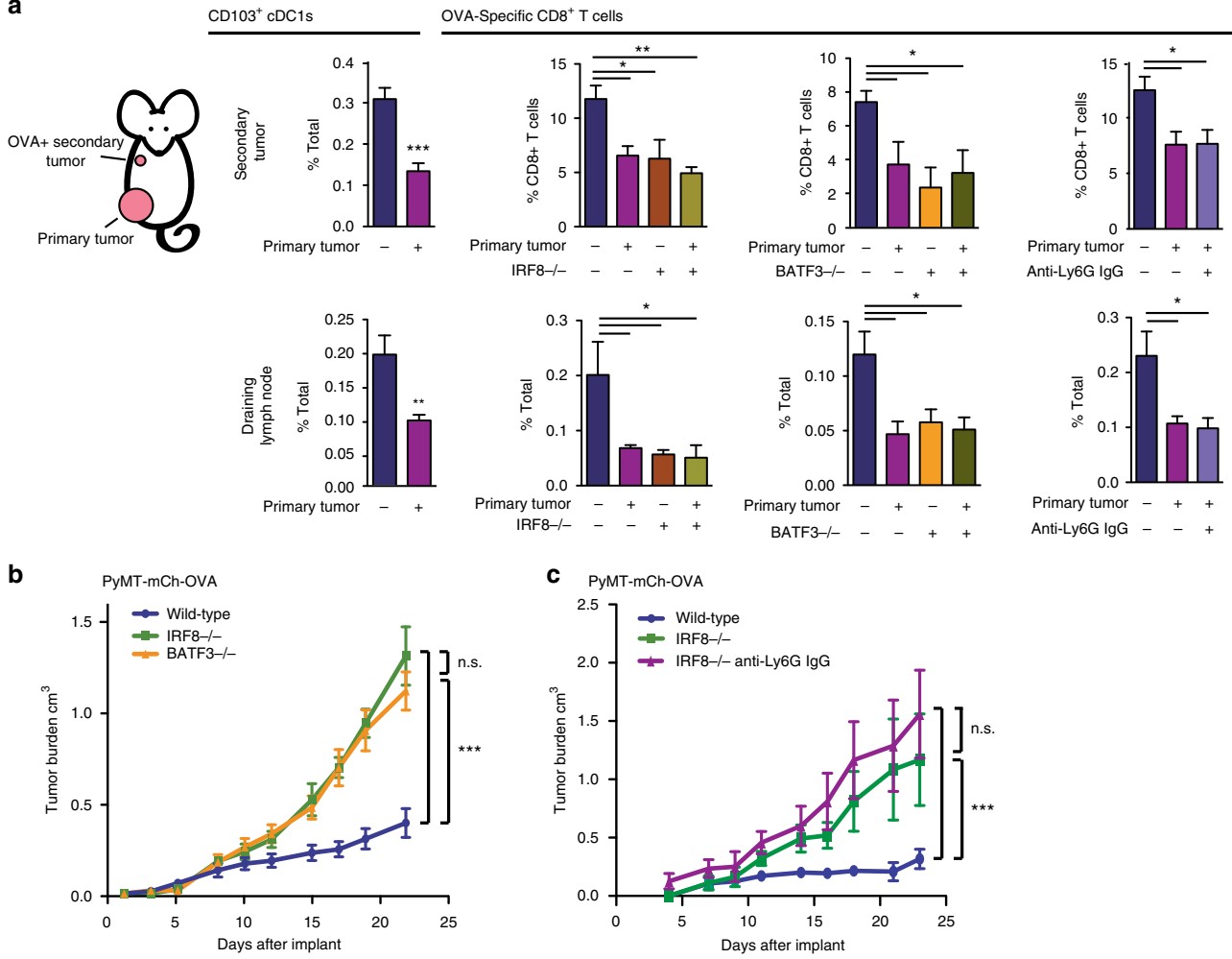

**Fig. 8** IRF8 and cDC1s are necessary for anti-tumor CD8+ T-cell responses. **a** Mice with 1.0 cm diameter orthotopic PyMT-B6 tumors and tumor-free controls were implanted with PyMT-B6-mCh-OVA cells in the upper mammary fat pad. After 7 days, the secondary tumor and its draining LN were analyzed for frequency of CD103+ cDC1s and OVA-specific CD8+ T cells. OVA-specific T-cell analysis was also performed in age-matched IRF8−/− and Batf3−/− mice and mice treated with anti-Ly6G IgGs starting 1 day before implant of upper mammary fat pad tumor; n = 5 mice/group. Experimental replicates are displayed. **b** Tumor volume over time in Batf3−/−, IRF8−/−, and wild-type controls implanted with orthotopic PyMT-B6-mCh-OVA tumors; wild-type, n = 10; BATF3−/−, n = 7; IRF8−/− n = 4. Data are representatives of two individual experiments. **c** Tumor volume over time in IRF8−/−, IRF8−/− treated with anti-Ly6G IgGs starting 1 day prior to implant and wild-type controls implanted with orthotopic PyMT-B6-mCh-OVA tumors; wild type, n = 6; IRF8−/−, n = 3; IRF8−/− treated with anti-Ly6G IgGs, n = 3. Error bars represent mean +/− s.e.m.; *p < 0.05, **p < 0.01, ***p < 0.001, n.s., not significant by unpaired two-sided Student's t test or two-way ANOVA

to the BM differentiation effect we have identified, reduced IRF8 expression is also detrimental to cDC maturation following activation signals in the periphery because IRF8 regulates a multitude of genes involved with antigen presentation and expression of interleukin 12 (IL-12)[19,54]. Together, these results suggest tumors can shift the net balance of immune-stimulatory and immune-suppressive BM and peripheral myeloid cells via alteration of IRF8 expression through regulation of inflammatory cytokines like GCSF, thereby blunting anti-tumor immunity.

Though granulocyte–monocyte expansion in BC and PDAC has been shown to impede tumor immune surveillance[7–9], it is important to consider the role of cDC1s in orchestrating anti-tumor CD8+ T-cell responses. In these studies, the presence of primary tumors is sufficient to reduce the pool of pre-cDC1s and interrupt tumor-specific CD8+ T-cell expansion. The number of CD8+ T cells in the tumor environment is important for response to both chemotherapy and immunotherapy[3,55–58]. cDC1s are known to regulate CD8+ T-cell numbers and function in the tumor environment. To this end, cDC1s cross-present tumor-

associated antigen to reactivate CD8+ T cells within tumor tissues and transport antigen to the draining LNs, where they stimulate naive T cells[2,6,18,20,22,23]. Intratumoral cDC1s also recruit T cells from the LNs into the tumor by expressing C-X-C motif chemokine ligand 9/10 (CXCL9/10)[21,26]. Furthermore, memory T-cell responses are deficient in mice depleted of cDC1s, suggesting cDC1s are important for re-challenge, which could manifest at the time of metastasis or tumor recurrence in patients[21]. Together, these functions identify cDC1s as critical supporters of anti-tumor CD8+ T cells. Given cDC1 differentiation is interrupted in cancer, our study shows that there is no longer a sufficient supply of cDC1 progenitors available to populate new and persistent sites of inflammation, such as sites of tumor outgrowth and metastasis, and that this lack of cDC1s is detrimental to anti-tumor CD8+ T-cell responses.

Through their function in supporting CD8+ T-cell responses, cDC1s are important for tumor control and response to therapies. In agreement with these data, others have shown that cDC1s are required for tumor control at both primary and metastatic

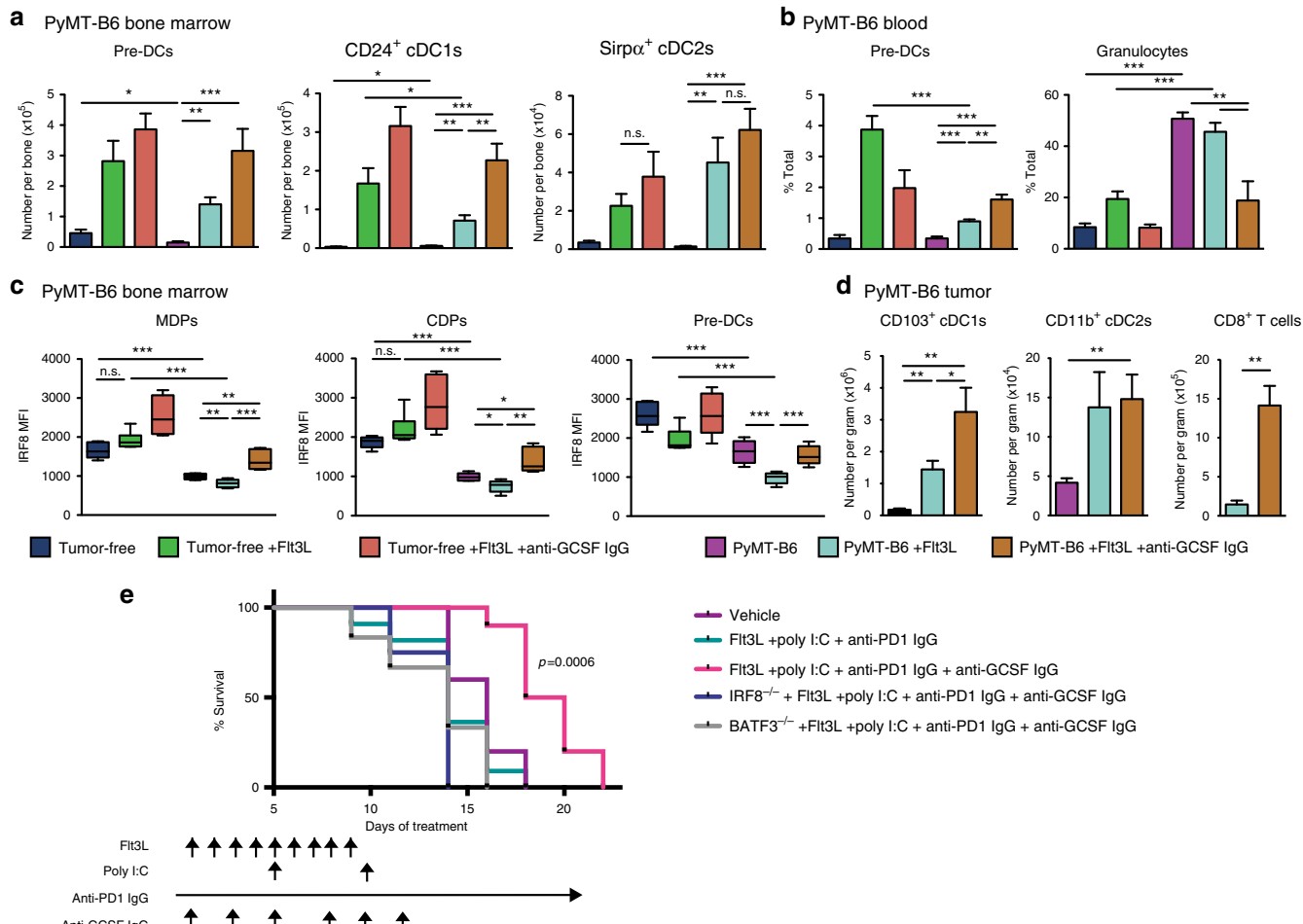

**Fig. 9** GCSF neutralization synergizes with Flt3L in vivo to promote cDC1 numbers and function. Tumor-free mice or orthotopic PyMT-B6 tumor-bearing end-stage mice treated with 50 μg anti-GCSF IgGs 3×/week for 2 weeks and/or 30 μg Flt3L daily for 9 days. End stage is defined in the Methods. **a** Number of BM pre-DCs, CD24$^+$ cDC1s, and Sirpα$^+$ cDC2s. **b** Number of blood pre-DCs and granulocytes. **c** IRF8 measured in BM MDPs, CDPs, and pre-DCs. **d** Number of tumor CD103$^+$ cDC1s, CD11b$^+$ cDC2s, and CD8$^+$ T cells (CD45$^+$CD3$^+$CD8$^+$); $n = 6$/group. **e** Established 1 cm+ diameter orthotopic PyMT-mCh-OVA mammary tumors were treated with vehicle or Flt3L (30 μg)+anti-PD1 IgGs (200 μg)+intratumoral Poly I:C (50 μg)+/− anti-GCSF IgGs (50 ng) according to the displayed treatment schedule. Survival to 2.3 cm$^3$ tumor volume. Error bars represent mean +/− s.e.m.; *$p < 0.05$, **$p < 0.01$, ***$p < 0.001$, n.s., not significant by unpaired two-sided Student's $t$ test. Log-rank (Mantel–Cox) $p$ value is denoted for differences in survival

sites[1,18,24,43] Furthermore, cDC1s are required for responses to checkpoint immunotherapy, and increased cDC1 numbers improve the response to chemotherapy in some cancer models[2,3,25,26,59]. Others have shown that the number of CD141$^+$ cDC1s in the tumor, especially in balance with immune-suppressive myeloid cells, is an important indicator of chemotherapy response and outcome in patients[1,3,8,36]. We found that increased BM CD141$^+$ cDC1 levels, in balance with reduced BM granulocytes, correlate with pCR in BC patients undergoing neoadjuvant chemotherapy. These data suggest tumor-induced alteration in myeloid differentiation, and specifically cDC1 development occurring in the BM, may also impact patient response to therapy and predict patient outcome. Reduced IRF8 expression in the tumor as a marker of cDC1s is correlated with worse patient outcome[1,3]. Extending upon this finding, we found reduced IRF8 expression in circulating pre-DCs also correlates with reduced overall and recurrence-free survival in pancreatic cancer patients. Given our new understanding of how tumors employ IRF8 downregulation to alter myeloid differentiation, it would be interesting to know if IRF8 expression in the BM or blood of patients could be a novel biomarker of a patient's immune status and/or response to chemotherapy and immunotherapy.

cDC1s are rare in the tumor microenvironment, and herein we showed that cDC1s are limited during development in the BM by tumor-induced inflammatory cytokines like GCSF[1,2,60]. To improve adaptive immune responses against the tumor, especially in the context of immunotherapies and chemotherapies, we should consider strategies to bolster cDC1 BM development. Others have also shown that Flt3L treatment or colony-stimulating factor 1 (CSF1) neutralization expands cDC1 numbers in the tumor and increases response to both chemotherapies and immunotherapies[2,3]. These strategies likely impact BM cDC1 differentiation and undermine this newly identified mechanism of tumor immune evasion. Here, we neutralized GCSF to increase BM cDC1 differentiation, which was able to further increase the efficacy of Flt3L by refining the IRF8-mediated cDC1 differentiation program. It is important to understand the interactions between cytokines and BM myeloid development so we can better modulate the systemic myeloid environment. These strategies could then be used in patients to support anti-tumor immunity and response to therapy.

In summary, we have shown that BC and PDAC alter the balance of immune-stimulatory cDC1s versus immune-suppressive myeloid cells by regulating IRF8 expression. This process leads to a favorable immune environment for tumor

progression. This mechanism reveals potential new biomarkers of immune response and targets for combination therapies.

## Methods

**Human bone marrow, peripheral blood and tumor samples**. BM and peripheral blood were obtained from patients diagnosed with locally advance or unresectable pancreatic ductal adenocarcinoma or clinical stage II/III breast cancer at the Barnes-Jewish Hospital (St. Louis, MO, USA) from 2004 to 2014 and were followed for recurrence and survival in a prospectively collected database. Samples were collected under informed consent in concordance with Institutional Review Board (IRB) approval (IRB protocol numbers 201102244, 201101961, 201108117). At the time of collection, these patients had received no prior cancer-related treatment. Healthy donor BM and blood were collected from cancer-free volunteers. Blood was collected into vacuum tubes containing heparin or ethylenediaminetetraacetic acid (EDTA) (BD Bioscience). Cells were isolated by ficoll-density centrifugation and frozen in fetal bovine serum with 5% dimethyl sulfoxide. BM from PDAC patients and BC cohort 1 patients was isolated as previously described[61,62]. BM from patients in BC cohort 2 was similarly obtained and stored but was not subjected to ficoll-density centrifugation. Breast tumor biopsies were from patients at the time of metastatic diagnosis under informed consent in concordance with IRB approval (IRB protocol number 201102394). Primary pancreatic adenocarcinoma tissues were collected during surgical resection and verified by standard pathology (IRB protocol number 201108117).

**Cell lines and constructs**. The PyMT-B6 murine mammary tumor cell line was derived from the mammary tumor tissue obtained from an end-stage MMTV-PyMT C57BL/6 mouse by our laboratory. The cell line was validated for *PyMT* expression; pan-keratin positivity; and Vimentin, smooth muscle actin, and CD45 negativity. A subset of this cell line was labeled with click beetle red luciferase-mCherry reporter and transduced to express OVA (PyMT-mCh-OVA). PyMT-B6-GCSFKO was made using the lentiCRISPR v2 vector. Virus was packaged using 293T cells and helper plasmids pCMV-DR8.2 and pCMV-VSVG. Csf3 gRNA sequences used were aggacgagaggccgttcccc, ctacaagctgtgtgtcaccccg, and ggagacggctcgccttgctc. Clones were selected and screened by sequencing for depletion of the *Csf3* gene. The 4T1-FL-GFP murine mammary tumor cell line was obtained from Dr. Katherine Weilbaecher's laboratory, originally from Dr. David Piwnica-Worms (Washington University in St. Louis, MO, USA). It had been previously modified to express firefly luciferase and green fluorescent protein, as previously described[63]. The Pan02 murine pancreatic tumor cell line was obtained from Dr. David C. Linehan (University of Rochester, NY, USA). The KP 1.0 murine pancreatic tumor cell line was derived from the pancreatic tumor tissue obtained by our laboratory from an end-stage KPC mouse and has been previously reported[64]. All cell lines were maintained in Dulbecco's modified Eagle's medium (DMEM) (Lonza) supplemented with 10% fetal bovine serum (Atlanta Biological) and penicillin/streptomycin (Gibco). All cell lines tested negative for mycoplasma.

**Genetic and orthotopic mouse models**. Mice were maintained in the Washington University Laboratory for Animal Care barrier facility. All studies were approved by the Washington University School of Medicine Institutional Animal Studies Committee. MMTV-PyMT FVB/N were obtained from The Jackson Laboratory and have been previously described[65]. Mice were analyzed at end stage, defined as approximately 3 months of age or when tumors reached >1.5 cm diameter. KPC (*p48-Cre;LSL-Kras^{G12D};Trp53^{flox/+}* C57BL/6) and KPPC (*p48-Cre;LSL-Kras^{G12D}; Trp53^{flox/flox}* C57BL/6) component mice were either obtained from The Jackson Laboratory (Kras and p53) or from Dr. Sunil Hingorani (p48, University of Washington, WA, USA) and have been previously described[64]. KPC experiments were performed with mice of mixed genders and mice were analyzed at end stage, defined as approximately 6 months of age or when tumors reached 1.0 cm, the animal experienced >15% weight loss, or other absolute survival event. For genetic models, aged matched littermates lacking oncogene expression were used as tumor-free controls. Zbtb46-GFP (B6.129S(C)-*Zbtb46^{tm1.1Kmm}*/J), IRF8^{−/−} (B6(Cg)-*Irf8^{tm1.2Hm}*/J) and Batf3^{−/−} (B6.129S-*Batf3^{tm1Kmm}*/J) mice in the C57BL/6 background were obtained from The Jackson Laboratory and have been previously described[18,29,66]. A total of 2.5 × 10^5 PyMT-B6, PyMT-mCh-OVA, or 4T1-FL-GFP cell lines were orthotopically implanted in low growth factor matrigel (Cultrex) into the mammary fat pad 4/5, while 2 × 10^5 Pan02 cells were orthotopically implanted in low growth factor matrigel (Cultrex) into the pancreas. All cell lines were implanted into C57BL/6 mice, except 4T1, which was implanted into BalbC mice. PDAC mouse models were analyzed at end stage, defined as tumors reaching 1.0 cm, the animal experienced >15% weight loss, or other absolute survival event. BC mouse models were analyzed at end stage, defined as tumors reaching 1.5 cm, the animal experienced >15% weight loss, or other absolute survival event. For secondary tumor experiments, 5 × 10^5 PyMT-mCh-OVA cells were implanted into mammary fat pad 2/3 when the 2.5 × 10^5 PyMT-B6 cells implanted into the mammary fat pad 4/5 produced a tumor 1.0 cm in diameter, or into mammary fat pad 2/3 of non-primary tumor-bearing matched controls. Mice were analyzed 1 week after secondary tumor implant. For anti-Ly6G IgG-treated arm, anti-Ly6G IgGs treatment, as described below, was started 1 day prior to implantation of the PyMT-mCh-OVA tumor. For poly I:C OVA-TxRd plug experiments, a 200 μl low

growth factor matrigel with 20 μg poly I:C and 40 μg OVA-TxRd Conjugate (ThermoFisher Scientific) was implanted into mammary fat pad 2/3 when 2.5 × 10^5 PyMT-B6 cells implanted into the mammary fat pad 4/5 produced a tumor 1.0 cm in diameter, or non-primary tumor-bearing matched controls. Mice were analyzed 10 days after the plug was implanted. Mice for implantation and controls were obtained from either Charles Rivers Laboratories or The Jackson Laboratories. For anti-Ly6G IgG-treated arm, anti-Ly6G IgGs, as described below, was started 1 day prior to implantation of the matrigel plug. For Ccl4-containing plug iteration, a matrigel plug containing 200 μl low growth factor matrigel with 20 μg poly I:C and 100 ng Ccl4 (PeproTech) was implanted and analyzed as described above. Tumor-bearing mice were randomized into treatment groups, when necessary. During randomization, animals were sorted by tumor size in ascending order, and then separated into groups in descending order. Investigator was blinded to treatment group during sorting. Groups were determined to have no statistical difference in average starting tumors size post hoc. Unless otherwise stated, animals were female C57BL/6 mice implanted with tumors at 7–8 weeks of age and analyzed at 9–12 weeks of age.

**Flow cytometry analysis**. For mouse tissue, single-cell suspensions were obtained prior to staining. BM was flushed from long bones for most analyses; for mature cDC staining, the long bone was crushed and digested in DMEM (Lonza) supplemented with 2 mg/ml collagenase A (Roche) for 25 min at 37 °C with agitation. For tumor tissue analysis, the mouse was subjected to perfusion with heparin-phosphate-buffered saline (PBS) (Alfa Aesar, Lonza) solution prior to tissue dissection. LNs, tumors, and matrigel plugs were minced and digested in DMEM (Lonza) supplemented with 2 mg/ml collagenase A (Roche) and DNAse (Sigma) for 25 min at 37 °C with agitation. All digestions were quenched with fetal bovine serum (Atlanta Biological). Blood was obtained by cardiac puncture and deposited in heparin-PBS (Alfa Aesar, Lonza) solution. Blood was then incubated in red blood cell lysis solution (Biolegend) for 10 min. All samples were washed with 1% bovine serum albumin (BSA) PBS (Sigma, Lonza) staining buffer and filtered with a 40 μM mesh filter (FisherBrand) prior to staining. Samples were blocked on ice with rat anti-mouse CD16/32 for 10 min (except when CD16/32-PE was used). Samples were then resuspended in 100 μl staining buffer with extracellular fluorophore-conjugated antibodies and incubated on ice for 30 min. Samples were washed with staining buffer and fixed with fixation buffer (BD) for 30 min. When intracellular staining was performed, the fixation/permeabilization kit (eBioscience) was used after extracellular staining according to the manufacturer's instructions. For green fluorescent protein (GFP) analysis, fixation was not performed and cells were analyzed immediately. Human BM and blood were thawed from cyro-preservation into PBS (Lonza). Live/Dead or viability dyes were applied for 15–30 min at room temperature (BD). Samples were then processed in the same manner as the mouse samples above. Data were acquired on the Fortessa X20, LSRII, or Fortessa system (BD). FlowJo v9 (Tree Star) was used for compensation and analysis. All antibodies and dilutions used are listed in Supplementary Tables 5 and 6. Markers for cell populations are listed in Supplementary Table 7.

**Microarray and RT-qPCR analysis**. Total RNA was isolated from GMP, MDP, and CDP (as defined in gating strategies) from the BM of end-stage PyMT-B6 tumor-bearing mice and tumor-free controls. Lineage depletion was performed using a Mouse Lineage Depletion Kit (Miltenyi Biotech) according to the manufacturer's instructions. Cells were processed and stained as described above and sorted on the ARIAII system (BD). RNA was isolated using the E.N.Z.A. Total RNA Kit (OMEGA) according to the manufacturer's instructions. Microarrays were performed on MDP and CDP samples by the Genome Technology Access Center (GTAC) at Washington University and data have been deposited in the National Center for Biotechnology Information (NCBI) Gene Expression Omnibus with accession number GSE99467. Differential gene list was generated and cluster analysis was performed using genes with detected fold changes >1.5, p < 0.05 and false discovery rate (FDR) q < 0.05. Microarray data were analyzed using DAVID Bioinformatics Resource to develop a list of Gene Ontology (GO) Terms[67,68]. RNA from GMPs, MDPs, and CDPs was also processed into complementary DNA (cDNA) using qScript cDNA SuperMix (Quantabio). Target genes were assessed using quantitative real-time PCR Taqman primer probes set for *Irf8*, *Cd209a*, *Zbtb46*, *Bcl11a*, *Spi1*, *SpiB*, *Cebpe*, *Gfi1*, *Cepba*, *Id2*, *Batf3*, *H2-Ke6*, *H2-Dma*, *H2-Q8*, *Tapbp*, *Tap2*, *Pml*, *Csf3r*, *Tbp*, and *Gapdh* (Applied Biosystems). Primer assay IDs are listed in Supplementary Table 8. Relative gene expression was determined on an ABI7900HT quantitative PCR machine (ABI Biosystems) using Taqman Gene Expression Master Mix (Applied Biosystems). The threshold cycle method was used to determine fold change gene expression normalized to *Gapdh* and *Tbp*. For in vitro gene expression assays, MPs, MDPs, and CDPs were sorted as described above. Cells were cultured in Rosewell Park Memorial Institute (RPMI)-1640 medium (Lonza) supplemented with 10% fetal bovine serum (Atlanta Biological), β-mercaptoethanol (Gibco), non-essential amino acids (Life Technologies), and L-glutamine (Life Technologies) in the presence of 100 ng/ml recombinant Flt3L and/or 100 ng/ml recombinant GCSF (PeproTech) for 24 h. Gene expression for *Irf8* was assessed by quantitative reverse transcriptase polymerase chain reaction (RT-qPCR) protocol described above.

**Immunohistochemistry (IHC) and in situ hybridization (ISH)**. Human PDAC and breast cancer tissue microarray (TMA) samples were stained using anti-GCSF IgGs (ab9691, Abcam) diluted 1:25 in blocking buffer (5% goat serum, 2.5% BSA in 1× PBS) on Bond Rxm autostainer. Antigen retrieval was performed using citrate-based Epitope Retrieval Solution (AR9961, Leica Biosystems), and immunostaining was visualized using 3,3′-diaminobenzidine by Bond Polymer Refine Detection Kit (DS9800, Leica Biosystems) using manufacturer's recommendations. Human PDAC and breast cancer TMA-stained slides were scanned at 10× magnification on Zeiss Axio Scan Z1 Brightfield/Fluorescence Slide Scanner, visualized and graded based on staining intensity. ISH was performed on mouse tumor and BM tissues isolated from end-stage PyMT-B6 and KPC mice and age-matched normal mammary glands, pancreas, and BM. Tissues were fixed in 10% formalin o/n at 4 °C and embedded in paraffin. Then, 6 μM sections were taken, dried overnight, and stained with ISH probe specific for CSF3 gene (Cat. No. 400918, ACD) using the RNAscope 2.5 LS Duplex Assay (Cat. No. 32240, ACD) using the manufacturer's recommendations for Leica Bond Rxm. For all quantifications, whole tissue slide scans were obtained at 10× or 20× magnification on Zeiss Axio Scan Z1 Brightfield/Fluorescence Slide Scanner. Additional 20× brightfield images were taken on the Nikon Eclipse 80i Epifluorescence microscope (Nikon). Whole tissue slide scans at 20× magnification were analyzed with HALO software (Indica Labs) using the Chromogenic RNA ISH Module. This allowed for thresholding and detection of positive probe staining on a single-cell basis to quantify average number of probe copies per cell.

**In vitro dendritic cell differentiation**. BM cells were isolated, lineage depleted, and stained as described above. CD45.2$^+$ MPs (Lin$^-$cKit$^+$ScaI$^-$), MDPs, or CDPs were isolated from whole BM of tumor-bearing or tumor-free mice by cell sorting on the ARIAII system (BD). A total of 2500 sorted progenitors were plated on 1.125 × 10$^6$ CD45.1$^+$ BM cell feeder culture in RPMI-1640 medium (Lonza) supplemented with 10% fetal bovine serum (Atlanta Biological), β-mercaptoethanol (Gibco), non-essential amino acids (Life Technologies), and L-glutamine (Life Technologies) in the presence of 100 ng/ml recombinant Flt3L and/or 100 ng/ml recombinant GCSF (PeproTech) in 24-well plates. The medium was replaced after 3 days and cultures were analyzed after 5 days. To lift cells, 0.05% Trypsin (HyClone) was used. Cells were stained and analyzed as described for flow cytometry, identifying the progeny of isolated progenitors by CD45.2 positivity. For GCSF priming experiment, progenitors were treated with 100 ng/ml GCSF or media alone for 24 h prior to plating in differentiation assay described above.

**IRF8 overexpression in vitro**. Retroviral vectors, MSCV Irf8 T2a Thy1.1, or MSCV empty vector T2a Thy1.1 control were obtained from Drs. Theresa and Kenneth Murphy (Washington University in Saint Louis, MO). Vectors were transfected into Phoenix-E cells using Trans-IT LT-1 (Mirus) according to the manufacturer's instructions. Virus was collected at 48 h after transfection and concentrated by spin at 25,000 rpms for 2 h. CD45.2$^+$Lin$^-$cKit$^+$ScaI$^-$MPs were sorted at previously described and infected with concentrated virus in 8 μg/ml polybrene by spin infection at 1800 rpms for 45 min. Infection was allowed to proceed overnight. Cells were then plated into dendritic cell differentiation assay described above. Cells were analyzed at 3 days. Cells were maintained in 100 ng/ml Flt3L and/or 100 ng/ml GCSF through infection and differentiation assay.

**In vivo differentiation**. BM CD45.1$^+$Lin$^-$cKit$^+$ScaI$^-$MPs were isolated as described above and transferred into sub-lethally irradiated 1.0 cm PyMT-B6 tumor-bearing mice and tumor-free controls by retro-orbital injection. After 2 weeks, CD45.1$^+$ populations were analyzed by flow cytometry as described above.

**MDP/CDP adoptive transfer**. BM MDP and CDPs were isolated as described above from the tumor-free mice and mice bearing end-stage orthotopic PyMT-B6 tumors. A total of 26,000 cells were adoptively transferred into each BATF3$^{-/-}$ mouse by retro-orbital injection. PyMT-mCh-OVA was implanted as described above 3 days after adoptive transfer.

**Cytokine analysis**. Blood serum was isolated from end-stage PyMT-B6 tumor-bearing mice and tumor-free controls. Blood was collected by cardiac puncture and allowed to clot for 30 min at room temperature. Clotted blood was separated by spinning at 1000 × g for 15 min. Serum cytokines were measured by Milliplex Multiplex Assay (EMD Millipore) according to the manufacturer's instructions. Assays were analyzed on the Luminex 100 (Luminex Corp.). Serum Flt3L was measured by Mouse Flt-3 Ligand DuoSet ELISA (R&D Systems) according to the manufacturer's instructions. Serum GCSF was measured by Mouse GCSF DuoSet ELISA (R&D Systems) according to the manufacturer's instructions. Human patient serum was analyzed for GCSF using a Human GCSF ELISA Kit (Abcam) according to the manufacturer's instructions.

**In vivo cytokine, antibody and drug treatment**. Mice were subcutaneously injected with 2 μg recombinant GCSF (PeproTech) in 100 μl PBS every day for 10 days. Mice bearing 1.0 cm diameter PyMT-B6 tumors were treated with 30 μg

Flt3L (Celldex) daily for 9 days by intraperitoneal injection. Mice bearing 1.0 cm diameter PyMT-B6 tumors were treated with 50 μg anti-GCSF IgGs (clone 67604, R&D Systems), 500 μg anti-IL-6 IgGs (clone MP5-20F3, BioXCell), or matched isotype control three times per week by intraperitoneal injection for 2 weeks. Mice were treated with anti-Ly6G IgGs (clone 1A8, BioXCell) or matched isotype control, first dose 400 μg, 100 μg for subsequent doses, three times per week beginning at time point indicated in the experiment. Mice were treated with 200 μg anti-PD1 IgGs (clone RMP1-14, BioXCell) every 3 days by intraperitoneal injection. Mice were treated with 50 μg poly I:C every 5 days by intratumoral injection, according to time points indicated in the experiment.

**Statistical analysis**. GraphPad Prism Version 5 (GraphPad Software Inc.) was used for statistical analyses. The sample size was determined for all human and mouse analyses and in vivo experiments using experimental data from other studies to estimate appropriate numbers of samples and mice to achieve >85% confidence for a twofold change in any given parameter at the $p < 0.05$ significance level. The number of animals and in vitro replicates is specified in the figure legends. Variance was analyzed using an $F$-test. Parametric data were compared using an unpaired $t$ test. Non-parametric data were compared using the Mann–Whitney $U$-test. The $p < 0.05$ was considered statistically significant. Data sets were tested for outliers using Grubbs' test (extreme studentized deviate method). For tumor burden studies, two-way analysis of variance (ANOVA) was used to compare groups and $p < 0.05$ was considered statistically significant. Spearman's correlation was used to compare human data points. For spearman's correlations, correlation coefficient ($r^2$) and $p$ values are reported in the figure. Log-rank (Mantel–Cox) tests were used to assess differences in survival and $p$ values are indicated in the figure; $*p < 0.05$; $**p < 0.01$; $***p < 0.001$; n.s. denotes not significant. All data were presented as mean $+/-$ s.e.m. or as box plots.

**Data availability**. Microarray data displayed in Fig. 3d and Sup. Figure 6b are deposited in NCBI Gene Expression Omnibus under accession code GSE99467. All other remaining data are available within the article and Supplementary Files, or available from the authors upon request.

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

## Acknowledgements

We would like to thank T. Murphy and K. Murphy (Washington University in St. Louis, USA) for the MSCV Irf8 T2a Thy1.1 and MSCV empty vector T2a Thy1.1 control as well as intellectual input. This work was supported by funding awarded to D.G.D. from the Department of Defense (W81XWH-15-1-0385) and the National Cancer Institute (P50 CA196510, R01 CA177670, R01 CA203890). We thank the Genome Technology Access Center in the Department of Genetics at Washington University School of Medicine for help with genomic analyses. The Center is partially supported by NCI Cancer Center Support (P30 CA91842) to the Siteman Cancer Center and by ICTS/CTSA (UL1RR024992) from the National Center for Research Resources (NCRR), a component of the National Institutes of Health (NIH), and NIH Roadmap for Medical Research.

## Author contributions

D.G.D. managed the project and coordinated author activities. M.A.M. designed and performed experiments including mouse modeling, flow cytometry analysis, and gene expression analysis. J.M.B. designed and performed experiments including mouse modeling, CRISPR CAS9, and tissue staining. B.L.K and J.M.B. supported experimental design and execution. X.S. and K.N.W. assisted in CRISPR CAS9 and provided intellectual input. G.A.C. performed adoptive transfer and irradiation experiments and supported data interpretation. T.M.N, W.G.H., R.C.F., D.C.L., C.M., K.N.W., and R.L.A. collected human patient samples and assisted in patient outcome analysis. R.Z.P. performed statistical analysis and support patient outcome analysis. R.F. assisted in acquisition of BC patient samples and assisted in patient data interpretation. R.L.A. provided intellectual input on the project. D.G.D. and M.A.M. wrote the manuscript with input from all authors.

## Additional information

**Competing interests:** The authors declare no competing interests.

