## [Peer Review File(PDF 191 kb) · Nature Communications]

REVIEWERS' COMMENTS:

Reviewer #2 (Remarks to the Author):

The authors have addressed my concerns productively, and also pointed out a limitation in the work of Salmon et al I had missed. Having looked at the responses to my fellow reviewers, I believe they should be largely satisfied as well. Given the potential importance of DC dynamics in defining the "immune desert" or "cold" immune phenotype found in many tumors, the authors' work gives us a new way of thinking about where key control elements may lie, even if certain aspects of the work are not entirely definitive. I believe this work is important and will be widely read. Publication in Nature Communications -- or even **** (which would be preferred in my view) -- is now clearly warranted.

Reviewer #3 (Remarks to the Author):

This manuscript is a revision of an original submission to ****, in which the authors have addressed the first round of critiques by additional experimentation and modifications to the text. These revisions significantly strengthen the proposed mechanism; namely, tumor-derived G-CSF inhibits cDC1-mediated anti-tumor immunity by blocking development of cDC1s from bone marrow progenitors. The major concerns of the prior review have been addressed adequately, and the study adds novel insight to our understanding of tumor-mediated immune suppression mechanisms. A few minor points remain:

The presentation of data within Supplementary Figs. 2 and 3 is out of sequence and thus a bit confusing in the text. Since Supplementary Fig. 2 shows gating strategies, this figure could be referenced early yet not repeatedly referenced along with data shown in Supplementary Fig. 3.

There are a few other cases in which data presentation is out of order, with information in numerically later Figures referenced before results presented in earlier Figures. It would be preferable for data presentation in the text to match the numerical Figure order.

On p. 8 of the text, the authors reference splenic granulocyte amounts, yet these are not shown in Supplementary Fig. 4a as indicated.

Reviewer #3 (on Reviewer#1 rebuttal):

Editorial note: Reviewer#3 expressed to the editor their satisfaction.

Reviewer #4 (Remarks to the Author):

The authors have done an excellent job to address the concerns of the Referees and to improve

the manuscript. The newly added data strengthen the conclusions of the revised manuscript. This manuscript offers new important insights into the impact of primary tumors on hematopoiesis, and subsequent DC development.